# NETosis and thrombosis in vaccine-induced immune thrombotic thrombocytopenia

Halina H. L. Leung [1,9], Jose Perdomo[1,9], Zohra Ahmadi[1], Shiying S. Zheng[1,2], Fairooj N. Rashid [3,4], Anoop Enjeti[5,6], Stephen B. Ting [7], James J. H. Chong [3,4,8] & Beng H. Chong [1,2] ✉

Vaccine-induced immune thrombotic thrombocytopenia (VITT) is a rare yet serious adverse effect of the adenoviral vector vaccines ChAdOx1 nCoV-19 (AstraZeneca) and Ad26.COV2.S (Janssen) against COVID-19. The mechanisms involved in clot formation and thrombocytopenia in VITT are yet to be fully determined. Here we show neutrophils undergoing NETosis and confirm expression markers of NETs in VITT patients. VITT antibodies directly stimulate neutrophils to release NETs and induce thrombus formation containing abundant platelets, neutrophils, fibrin, extracellular DNA and citrullinated histone H3 in a flow microfluidics system and in vivo. Inhibition of NETosis prevents VITT-induced thrombosis in mice but not thrombocytopenia. In contrast, in vivo blockage of FcγRIIa abrogates both thrombosis and thrombocytopenia suggesting these are distinct processes. Our findings indicate that anti-PF4 antibodies activate blood cells via FcγRIIa and are responsible for thrombosis and thrombocytopenia in VITT. Future development of NETosis and FcγRIIa inhibitors are needed to treat VITT and similar immune thrombotic thrombocytopenia conditions more effectively, leading to better patient outcomes.

Vaccine-induced immune thrombotic thrombocytopenia (VITT), also known as thrombosis with thrombocytopenia syndrome (TTS), is an uncommon but serious adverse effect of adenoviral vector-based SARS-CoV-2 (COVID-19) vaccines, specifically ChAdOx1 nCoV-19 (Vaxzevria, AstraZeneca) and Ad26.COV2.S (Janssen/Johnson & Johnson)[1,2]. VITT resembles heparin-induced thrombocytopenia (HIT) which is an immune reaction to a commonly used anticoagulant, heparin[3]. Like patients with HIT, patients with VITT present with thrombocytopenia (low platelets) and thrombosis (blood clots, often at unusual sites) and have an anti-platelet factor 4 (PF4) antibody which induces platelet activation[4]. The high mortality of VITT (fatality rate estimated at 23%[5] to 40%[4]) has caused serious concerns among physicians, public health officials and the public, leading to vaccine hesitancy and undermining vaccine roll-out in many countries. This is exacerbated by the lack of knowledge of its underlying disease mechanism.

However, the seriousness of VITT from a risk-benefit perspective of COVID-19 vaccines is of relevance[6]. The risk of developing VITT is small (1 case per 26 500 to 127,300 after the first vaccine dose[7]), while the risk of COVID-19 infection is still highly significant worldwide, having caused millions of deaths and huge morbidity, and social economic damage around the globe.

VITT was identified after the widespread use of COVID-19 vaccines[1,2]. Concepts of disease pathogenesis are evolving and some require further scientific research and validation. It is generally

[1]Haematology Research Unit, School of Clinical Medicine, St George and Sutherland Campus, Faculty of Medicine and Health, University of New South Wales, Sydney, NSW, Australia. [2]New South Wales Health Pathology, Sydney, NSW, Australia. [3]Sydney Medical School, Faculty of Medicine and Health, University of Sydney, Sydney, NSW, Australia. [4]Centre for Heart Research, Westmead Institute for Medical Research, University of Sydney, Sydney, NSW, Australia. [5]Calvary Mater Hospital, Wallsend, NSW, Australia. [6]University of Newcastle, Callaghan, NSW, Australia. [7]Department of Haematology, Eastern Health and Monash University, Melbourne, VIC, Australia. [8]Department of Cardiology, Westmead Hospital, Sydney, NSW, Australia. [9]These authors contributed equally: Halina H.L. Leung, Jose Perdomo. ✉e-mail: beng.chong@unsw.edu.au

believed that platelet activation by the anti-PF4 antibody causes thrombosis in VITT despite the lack of scientific evidence of this antibody inducing clot formation either in vitro or in vivo. Recent studies have found that platelet FcγRIIa and NETosis are involved in VITT-induced thrombosis[8,9], however further investigations are needed to dissect the respective biological mechanisms leading to thrombosis and thrombocytopenia in VITT. Experts have suggested that there is a need to show in vivo thrombus formation by the anti-PF4 antibody in a VITT animal model[4] and also to understand the mechanism that causes thrombosis.

In HIT, we have previously shown that thrombosis is driven by NETosis[10,11]. Upon activation by pathogens, immune complexes and other stimuli, neutrophils release their granules and decondensed chromatin in the form of a DNA network, termed Neutrophil Extracellular Traps (NETs). This process is known as NETosis. Two characteristic components of NETs, myeloperoxidase and citrullinated histone H3 (CitH3) are often used as markers of NETs formation. NETs serve as a framework for thrombus formation and are highly thrombogenic – they activate platelets and other immune cells, damage endothelial cells[12] and activate blood coagulation pathways[13]. NETosis is known to promote venous and arterial thrombosis[14,15]. NETs have also been detected in patients with COVID-19 infection[16].

In this report, using a microfluidics blood flow assay we show that the anti-PF4 antibody (purified IgG from patients with VITT) when added to circulating normal whole blood induced blood clot formation in vitro and when administered into FcγRIIa[+]/hPF4[+] transgenic mice (a VITT animal model) induced thrombosis in vivo. We also demonstrated that the antibody-induced thrombosis was mediated by platelet and neutrophil activation and NETosis.

## Results

### VITT patients
VITT patients ($n = 7$) from five hospitals in Australia participated in the study. Mean age was 64 years (range: 46–84 years), three were female. Their clinical features and laboratory test results (Supplementary Table 1) are consistent with those of previously reported cases of VITT[2,5,17]. All received their first dose of COVID-19 vaccine (Vaxzevria, AstraZeneca) 12–32 days (mean: 19 days) before their admission to the hospitals, and blood samples collected soon thereafter. All had thrombocytopenia (mean platelet count at admission: $64 \times 10^9$/L, range: $8$–$138 \times 10^9$/L) and thrombosis (cerebral venous sinus thrombosis, CVST: two patients, splanchnic vein thrombosis: 2, bilateral pulmonary thromboembolism and deep vein thrombosis: 3). All had elevated D-dimer levels, reduced or normal plasma fibrinogen, anti-PF4 antibodies detected by enzyme-linked immunosorbent assay (Fig. 1a) and were positive for platelet activation functional assays (Fig. 1b, c). VITT data were compared with control groups which include (a) healthy individuals who received ChAdOx1 nCoV-19 vaccine within approximately the same time intervals as the VITT patients but did not develop VITT (denoted as Vax in the figures); (b) patients who had venous thromboembolism (denoted as VTE); and (c) patients with critical illness such as severe sepsis who were treated in intensive care units (denoted as ICU).

### NETosis in VITT
We next investigated the presence of markers of NETosis in VITT patients' plasma and fresh whole blood from patients with active VITT. We assessed both the presence of citrullinated histone H3 (CitH3)[18], myeloperoxidase (MPO) and the concentration of cell-free DNA (cfDNA) in plasma. The levels of CitH3, MPO and cfDNA were significantly increased relative to control groups (Fig. 1d–f), which is consistent with the presence of NETosis[11,19]. Moreover, analysis of fresh blood from patients with active VITT (using the gating strategy shown in Supplementary Fig. 1a–d), showed the presence of abundant activated neutrophils (low density granulocytes, LDG) (Fig. 1g, h),

neutrophil-platelet aggregates (NPA) (Fig. 1i, j) and neutrophils and LDG undergoing NETosis (Fig. 1k–m). Overall, this suggests that NETosis is present in patients with active VITT.

### VITT IgG induces NETosis in vitro
Pathogenic anti-PF4 antibodies bind to endogenous PF4 and form immune complexes[20]. These complexes interact with FcγRIIa to activate platelets and neutrophils[11,20]. To determine the effect of VITT antibodies in thrombosis, we first isolated total IgG from VITT patients' plasma and assessed its impact on whole blood from healthy donors. Compared to buffer (PBS) and control IgGs (Vax, VTE) incubation with VITT IgG led to a pronounced increase in the formation of LDG (Fig. 2a) and induction of neutrophils to undergo NETosis (CD15[+]CitH3[+]MPO[+] cells) (Fig. 2b). NETs induction in the absence of other cells was corroborated by treatment of purified neutrophils in the presence of PF4 with VITT or control IgG (Vax) and assessment of DNA release with the cell impermeant dye Sytox green. Significantly increased DNA release was triggered by VITT IgG relative to control IgG (Fig. 2c, d), indicating that in the presence of PF4, purified IgG from VITT patients strongly initiated NETs formation in healthy donors' whole blood and purified neutrophils in vitro. Monocytes are also known to generate ETs[21]. We found that purified monocytes were activated at low levels by VITT IgG in the presence of PF4 to induce monocyte ETs (Supplementary Fig. 1e, f). The contribution of monocytes to overall ETs is minimal due to their small numbers in circulation and lower reactivity to VITT IgG compared to neutrophils. Altogether, these data suggest that VITT IgG induces both neutrophils and monocytes to undergo ETosis and ETs in VITT are predominantly derived from neutrophils.

### Anti-PF4 VITT antibodies activate platelets and neutrophils
To exclude potential artifacts activating cells such as aggregated IgG in the patient's antibody preparation, the aggregation status of the purified total IgG was determined by size exclusion chromatography. As expected, 95.7% (range 92.2–98.1%) of the IgG preparations (VITT patients and control IgGs) were monomers (Supplementary Fig. 2a), indicating that the effects observed are due to monomeric VITT IgG activity and not to aggregated IgG (known to activate platelets and neutrophils). In support of this, control IgGs containing similar amounts of monomeric and aggregated IgG did not induce platelet and neutrophil activation.

To determine the specific antibody within the total IgG population responsible for inducing platelet and neutrophil activation, specific anti-PF4 IgG was affinity purified with biotinylated PF4 and its abundance and activity determined by gel electrophoresis and platelet/neutrophil activation assays, respectively. The anti-PF4 antibody was 0.6% (range 0.1–1%) of the total IgG (Supplementary Fig. 2b). The binding activity of affinity purified anti-PF4 IgG was confirmed by PF4 ELISA (Supplementary Fig. 2c). Affinity purified anti-PF4 was found to strongly activate platelets and induce NETosis at low concentrations, as analysed by serotonin release assay (SRA, 15 μg/mL, Supplementary Fig. 2d) and using purified neutrophils by flow cytometry (250 μg/mL, lower concentrations were not tested, Supplementary Fig. 2e).

### VITT IgG induces thrombosis in vitro
To examine the capacity of VITT antibodies to induce thrombosis, fresh whole blood from healthy donors was treated with VITT IgG or control IgG and flowed through von Willebrand factor (vWf)-coated microchannels in a microfluidics system. The presence of VITT IgG led to thrombus formation. Confocal microscopy imaging of thrombi formed following treatment with VITT IgG showed that the thrombi were formed by platelets, neutrophils and extracellular DNA, while no clots were formed in blood treated with control IgG (Supplementary Fig. 3a, b). Further analysis of VITT IgG-induced thrombi showed an abundance of fibrin (Fig. 2e) and CitH3 (Fig. 2f) confirming the strong thrombogenic activity of VITT antibodies and their ability to induce

# Platelet activation and NETosis in VITT

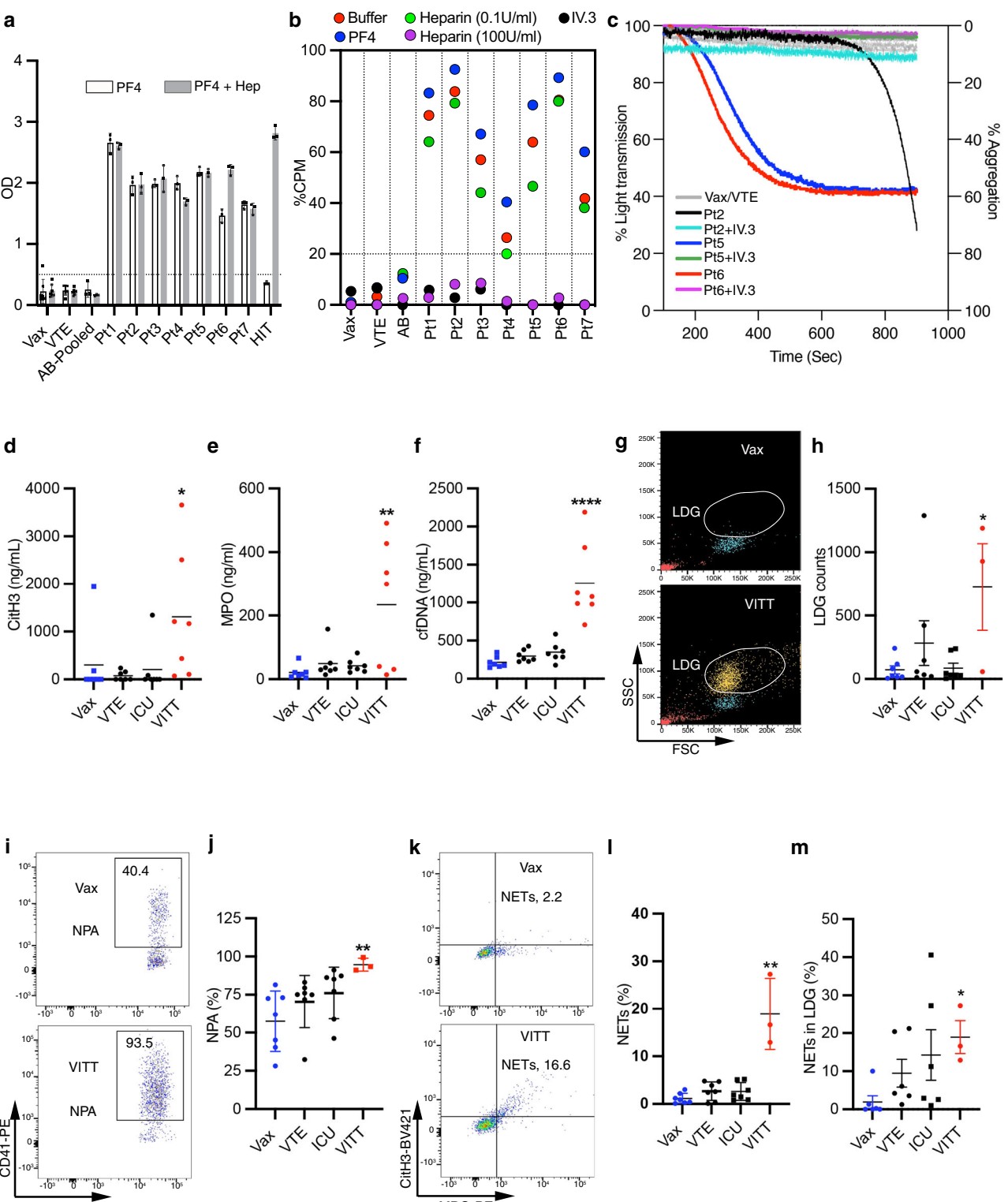

NETosis in vitro. To confirm the role of FcγRIIa and NETosis in VITT-induced thrombosis, blood was pre-treated with anti-FcγRIIa monoclonal antibody, IV.3 or DNase I prior to incubation with VITT IgG. There was no induction of NETosis as indicated by the absence of DNA release in the presence of IV.3 (Fig. 2g, h). Furthermore, IV.3 strongly inhibited deposition of platelets (Fig. 2g, i) and neutrophils (Fig. 2g, j). Similarly, the presence of DNase I resulted in inhibition of thrombus

formation (Fig. 2g–j). These data suggest that direct blocking of FcγRIIa inhibits NETosis and thrombosis, and digestion of extracellular DNA also inhibits thrombus formation in vitro.

## VITT IgG induces thrombosis in vivo

To assess whether anti-PF4 antibodies are responsible for the clinical features of thrombocytopenia and thrombosis in VITT patients, we

**Fig. 1 | Platelet activation and NETosis in VITT. a** PF4 and PF4-heparin ELISA experiment of VITT serum and controls. The cut-off, 0.50 OD units (Vax, VTE $n = 7$; AB, VITT Pt and HIT $n = 3$). **b** $^{14}$C-serotonin release assay for VITT samples with buffer alone, PF4 (10 μg/mL), 0.1 or 100 U/mL heparin or IV.3 antibody (50 μg/mL). Each dot represents the mean of assays done in triplicate. The cut-off was set at 20% CPM. **c** Platelet aggregation responses. Purified IgG from VITT patients induced aggregation in platelet-rich plasma (red, blue and black traces). Blockage of FcγRIIa with IV.3 inhibited aggregation (purple, green and light blue traces). **d** Nucleosomal CitH3 (H3R8Cit ELISA, $*p = 0.03$), **e** myeloperoxidase (ELISA, $**p = 0.01$), and **f** cfDNA (PicoGreen fluorescence assay, $****p < 0.0001$) levels in VITT patients' plasma ($n = 7$) relative to controls ($n = 7$) were determined. **g** Representative side and forward scatter flow cytometry plot backgated for neutrophils (yellow) and monocytes (blue) from VITT patient's and vaccine control blood. LDG are indicated. **h** LDG quantitated as number of LDG events relative to 200 monocytes ($n = 7$,

except VITT $n = 3$; $*p = 0.02$). **i** Representative plot of NPA from VITT and vaccine control blood. **j** Quantification of NPA in VITT ($n = 7$, except VITT $n = 3$; $**p = 0.004$). **k** Representative plot of NETs from VITT and vaccine control blood. Quantification of NETs in VITT in **l** whole blood ($n = 7$, except VITT $n = 3$; $**p = 0.005$) and in **m** LDG population ($n = 6$, except VITT $n = 3$; $*p = 0.02$). MPO$^+$, CitH3$^+$ double positive cells within the CD15$^+$ population were defined as neutrophils undergoing NETosis. The percentage of gated events is indicated in each quadrant. Statistics: Kruskal--Wallis ANOVA with Dunn's correction. Data are presented as **a**, **j**, **l**, **m** mean ± SD; **h** mean ± SEM. OD optical density units, CPM counts per minute, Vax ctrl healthy vaccinated subject IgG; VTE venous thromboembolism patient IgG, ICU intensive care unit patient IgG, NPA neutrophil-platelet aggregates, LDG low density granulocytes, cfDNA cell-free DNA, CitH3 citrullinated histone H3, Pt patient. Source data are provided in the Source Data file.

used a FcγRIIa$^+$/hPF4$^+$ double transgenic mouse model. These mice are necessary to assess the activity of VITT IgG in vivo, since they express two essential components, human PF4 (Supplementary Fig. 5a) and FcγRIIa on platelets and neutrophils[11]. VITT IgG was administered into the VITT mouse model and lungs extracted to examine the levels of thrombosis. Examination of extracted lungs (Fig. 3a, Supplementary Fig. 3c) from VITT IgG-treated mice showed extensive thrombi deposition in this organ. Quantitative analysis of thrombi in each treatment group confirmed a significant increase in clots in the lung of VITT mice compared to control groups (Supplementary Fig. 4a). These clots contain abundant platelets, neutrophils (Fig. 3b, c) and fibrin (Fig. 3c). Importantly, neutrophils within the lungs were positive for CitH3 (Fig. 3b, white arrows) suggesting that VITT antibodies activated neutrophils to undergo NETosis in vivo. Thrombi were absent in control IgGs (normal, Vax and VTE) treated animals (Fig. 4a, b). These data suggest that VITT IgG is responsible for thrombosis and NETosis in vivo.

### Role of FcγRIIa and NETosis in thrombosis in vivo
To investigate the role of FcγRIIa and neutrophil activation and NETosis in VITT IgG-induced thrombosis, inhibitors of FcγRIIa (aglycosylated IV.3[11], agIV.3) and NETosis (GSK484)[11,22] were administered in vivo. In support of our in vitro findings (Fig. 2) blocking either FcγRIIa or NETosis was effective in preventing the formation of clots in vivo as shown by the lack of clots (platelet and neutrophil accumulation) in lung sections of mice treated with agIV.3 (Fig. 3c). The dramatic reduction in thrombus deposition is also confirmed using whole organ imaging (Fig. 4a, b) and quantitative analysis of platelet accumulation in mice treated with VITT IgG plus agIV.3 or GSK484 compared to mice treated with VITT IgG without either inhibitor (Fig. 4a, b). Moreover, inhibitor-treated mice were not only protected from thrombosis but also had significantly less LDG present in peripheral blood compared to VITT IgG-treated mice (Supplementary Fig. 4b).

The contribution of NETosis to thrombosis in VITT was further assessed using the VITT mouse model deficient in PAD4 (FcγRIIa$^+$/hPF4$^+$/PAD4$^{-/-}$). PAD4 is the enzyme responsible for the citrullination of histones necessary for induction of NETosis[23]. Consistent with findings in animals treated with VITT IgG plus GSK484, PAD4$^{-/-}$ mice treated with VITT IgG had a dramatic reduction in clot formation compared to control VITT mice (which are wild type for PAD4) (Figs. 3c, 4a, b). There were also significantly fewer circulating LDG in PAD4 deficient mice compared to control (Supplementary Fig. 4b). Collectively, our data indicate that inhibition of platelet and neutrophil activation by blocking FcγRIIa or inhibition of NETosis can efficiently abolish VITT IgG-induced thrombosis in vivo.

### VITT IgG induces thrombocytopenia in VITT mouse model
Unlike mice treated with control IgGs (normal, Vax, VTE), mice treated with VITT IgG experienced thrombocytopenia (Fig. 4c, d) and systemic

reactions such as hypothermia (Supplementary Fig. 4c). AgIV.3 was effective in preventing both thrombocytopenia (Fig. 4d) and thrombosis (Figs. 3c, 4a). In contrast, NETosis inhibitor GSK484 and PAD4 knock-out had no effect on the development of thrombocytopenia (Fig. 4e) although they strongly inhibited thrombosis (Fig. 4a, b). Altogether, these results indicate that VITT IgG-induced thrombosis and thrombocytopenia are distinct pathobiological processes (Supplementary Fig. 5b).

## Discussion
Although vaccines against COVID-19 infection are very effective, there have been some serious side-effects. This has generated much public concern globally resulting in vaccine hesistancy and undermining of vaccine roll-out in many jurisdictions. These concerns should be taken in perspective given a higher risk of contracting COVID-19 where large vessel and microvascular thromboses are frequent and contribute to severe morbidity and mortality[24–26]. Despite the health risks involved in developing VITT, COVID-19 vaccines are needed to protect the community. Overall, the substantial benefit of the COVID-19 vaccines outweighs the small risk of VITT.

Despite numerous recent publications[1,8,27–29], there are still significant knowledge gaps in VITT, in particular regarding its underlying disease mechanism(s)[4]. The conventional concept is that the platelet activating anti-PF4 antibody causes clot formation in VITT despite no direct evidence. The presence of anti-PF4 antibodies in individuals without thrombosis[30] has created doubts about this concept. As in heparin treatment where non-pathogenic anti-PF4 antibodies are commonly found[31] non-activating anti-PF4 antibodies are more frequently found following COVID-19 vaccinations (both mRNA- and adenovirus-based vaccines)[30]. Conditions like VITT and HIT are rare and only develop in a very small subset of patients who produce pathogenic anti-PF4 antibodies.

The involvement of platelet FcγRIIa and NETosis in VITT has been reported recently[8,9,32]. There is yet no experimental evidence that shows VITT antibodies cause thrombosis and thrombocytopenia in vivo. This study addresses these research gaps and contributes to the understanding of the mechanism of VITT by establishing the first mouse model of VITT.

Here we provide evidence that VITT antibodies directly induce thrombus formation in vitro and in vivo, not by platelet activation alone but also through neutrophil activation and NETosis. In this study, we demonstrated that VITT IgG stimulated platelet activation via serotonin release and platelet aggregation assays as had other investigators previously[33]. Moreover, we showed that VITT IgG when added in vitro to circulating whole blood induced clot formation in the microchannels of a microfluidics system. In contrast, control IgG failed to induce thrombosis. Similarly, administration of VITT IgG but not control IgGs led to development of multiple thrombi in the lungs of the VITT mouse model (FcγRIIa$^+$/hPF4$^+$ double transgenic mice). These data provide direct evidence that VITT IgG (or more specifically

## Effect of VITT IgG on donor's blood

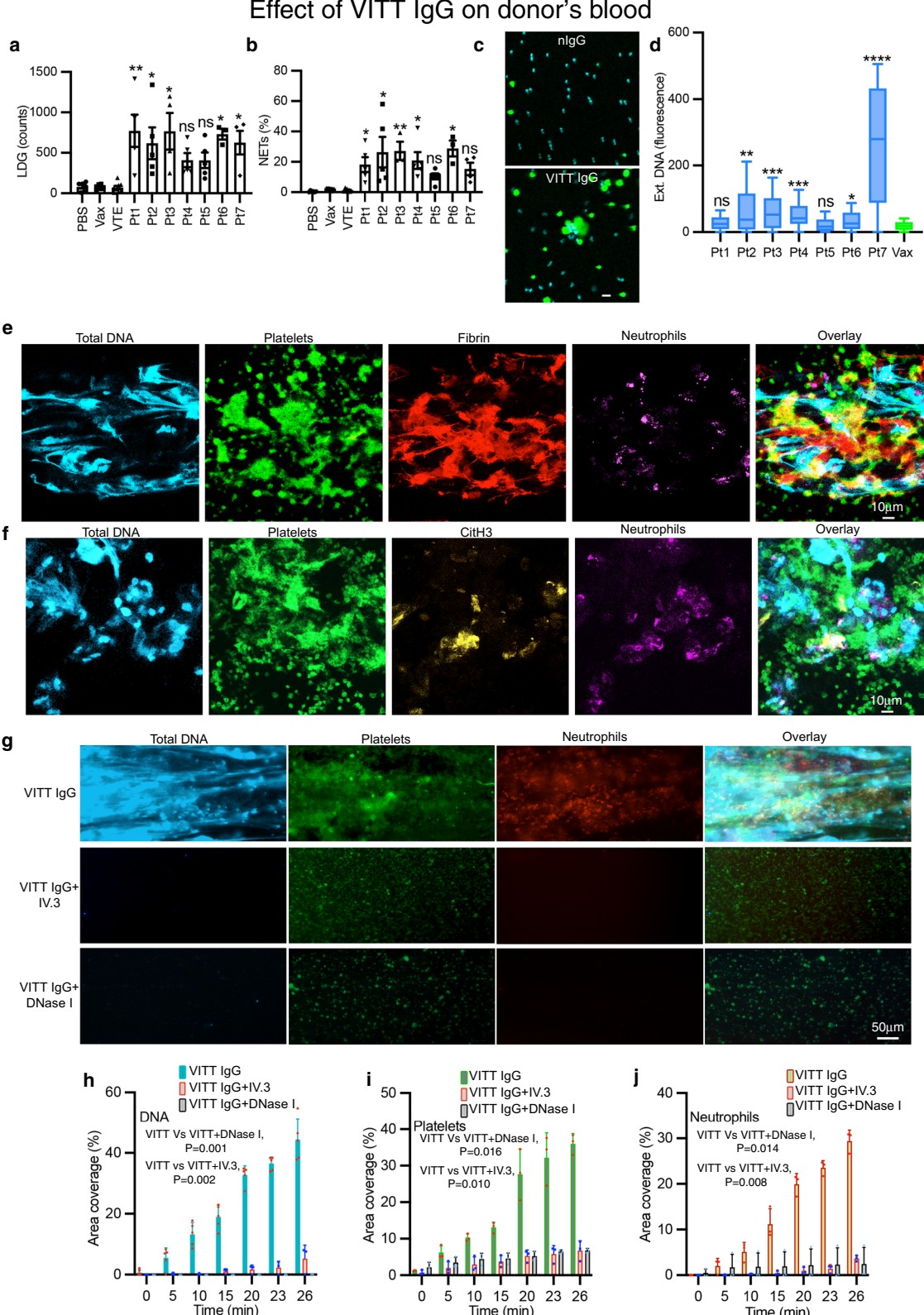

immune complexes formed by VITT IgG and PF4) induced clot formation in vitro and in vivo, filling a crucial knowledge gap in VITT pathogenesis. Thrombosis was blocked by anti-FcγRIIa monoclonal antibody, IV.3 suggesting that it was mediated by FcγRIIa receptors on platelets and neutrophils.

Our findings support recent studies showing the presence of NETosis in patients with VITT[8,9]. Holm et al.[8] showed that NETosis

markers occurred together with numerous markers of inflammation, activated innate immune pathways, activated blood cells and endothelium, and damaged tissues in VITT patients. These markers were present in the blood, in a VITT thrombus and in the immune precipitates extracted by a goat anti-human PF4 antibody from plasma of VITT patients[8]. These findings were not unexpected as the VITT patients had robust immune responses, intense inflammation and

**Fig. 2 | Effect of VITT IgG on donor's blood. a** Quantification of LDG (PBS n = 6; Vax n = 6; VTE n = 8; Pt1 n = 5, **p = 0.005; Pt2 n = 5, *p = 0.048; Pt3 n = 4, *p = 0.015; Pt4 n = 5, ns; Pt5 n = 5, ns; Pt6 n = 3, *p = 0.013; Pt7 n = 4, *p = 0.023) and **b** NETs following treatment of healthy donor blood with VITT IgG and controls (PBS n = 6; Vax n = 7; VTE n = 8; Pt1 n = 5, *p = 0.0383; Pt2 n = 5, *p = 0.0190; Pt3 n = 3, **p = 0.0057; Pt4 n = 5, *p = 0.0131; Pt5 n = 5, ns; Pt6 n = 3, *p = 0.0048; Pt7 n = 4, ns). **c** Purified neutrophils treated with VITT IgG or control IgG plus PF4 were stained for extracellular DNA (green) and nuclei (blue). Scale bar: 10 µm. **d** DNA release calculated as fluorescence intensity ratio of extracellular DNA (Sytox staining)/total DNA (Hoechst staining). The box extends from the 25th to 75th percentiles; the whiskers represent minimum (lowest 25%) and maximum (highest 25%) values; the line in the box represents the center of the data (median) (*p = 0.005, **p = 0.002, Pt3 ***p = 0.0002, Pt4 ***p = 0.0001, ****p < 0.0001; Ctrls n = 19; Pt1-7 n = 3). **e** VITT IgG induces thrombosis. Healthy donors' blood treated with VITT IgG was stained for total DNA (blue), platelets (green), fibrin (red) and neutrophils (magenta).

Thrombi were imaged with a confocal laser-scanning microscope (overlap of green and red shown as yellow). Scale bar: 10 µm. **f** Thrombi contain CitH3. Thrombi were generated and imaged as in **e**, and stained for DNA (blue), platelets (green), CitH3 (yellow) and neutrophils (magenta). Overlap of yellow and green is shown as white. **g** IV.3 and DNase I prevent VITT IgG-induced thrombus formation in microfluidics system. Treated blood was stained for DNA (blue), platelets (green) and neutrophils (red). Scale bar: 50 µm. Graphs show area coverage percentage for **h** total DNA (n = 3, except VITT IgG+DNase I n = 5), **i** platelets (n = 3) and **j** neutrophils (n = 3). Data presented as (**a, b**) mean ± SEM; **h–j** mean ± SD. Experiment was repeated independently (**c**) at least 10 times; **e–g** at least 3 times. Statistics: **a, b** Kruskal–Wallis test with Dunn's correction for multiple comparison, **d** Kruskal–Wallis test, **h–j** one-way ANOVA with Tukey's correction for multiple comparisons. Vax healthy vaccinated subject IgG, VTE venous thromboembolism patient IgG, LDG low density granulocytes, ext. DNA extracellular DNA, Ctrl control, Pt patient. Source data are provided in the Source Data file.

severe thromboses. However, there was no data implicating NETosis as the cause of thrombosis in the VITT patients. Even the presence of neutrophils and NETosis markers in the thrombus does not necessary indicate that it is the cause of thrombosis as neutrophils and NETs are frequently observed in thrombi in various conditions including stroke, acute myocardial infarction[34,35] and deep vein thrombosis[36]. Holm et al.[8] speculated whether the adenovirus in the vaccine or even the spike protein could have triggered the pronounced inflammatory processes including NETosis. Interestingly, Greinacher et al. found that VITT patient serum induced NETosis only in the presence of platelets suggesting that platelets play a key role in thrombosis in VITT[9].

In contrast, our study not only provides evidence of the presence of NETosis in VITT, but we also show that NETosis directly drives thrombosis in VITT in vivo in the VITT animal model. Administration of VITT IgG but not control IgGs induced development of multiple thrombi in the lungs of the mice. Thrombosis could be prevented or substantially suppressed by administration of NETosis inhibitor, GSK484 or by using PAD4 knock-out mice (which blocks NETosis). Furthermore, we found that neutrophils could be directly activated by VITT IgG and PF4 in the absence of platelets.

We also demonstrated that VITT antibodies induced thrombocytopenia in the VITT mouse model by binding to platelet FcγRIIa. Thrombocytopenia was substantially prevented by anti-FcγRIIa monoclonal antibody, IV.3. In contrast, NETosis inhibitor GSK484 and absence of PAD4 (FcγRIIa+/hPF4+/PAD4−/− mice), which significantly blocked thrombosis in VITT, had no effect on thrombocytopenia, suggesting that thrombosis and thrombocytopenia in VITT are two distinct processes as we have previously shown in heparin-induced thrombocytopenia[11].

Pathway-specific NETosis mechanisms are agonist-specific[37,38]. The requirement or dispensability of dinucleotide phosphate (NADPH) oxidase (NOX) activity, reactive oxygen species (ROS), and calcium influx on VITT IgG-induced NETosis requires further research. We have previously shown that PAD4, ROS and NOX2 play a crucial role in NETosis and thrombosis in HIT[10]. Due to the analogous nature of HIT and VITT, NET formation is likely dependent on both PAD4 and NADPH oxidase, as in HIT.

In summary, our findings show anti-PF4 antibodies are the pathogenic or disease-causing antibodies in VITT. They induce platelet and neutrophil activation leading to development of NETosis which is the major driver of thrombosis in VITT (Supplementary Fig. 5b). FcγRIIA blockage prevented both thrombocytopenia and thrombosis but NETosis inhibition which effectively suppressed thrombosis, had no effect on thrombocytopenia. Therefore, thrombosis and thrombocytopenia appeared to be mediated by two independent pathological mechanisms. Thrombocytopenia is likely to be due to direct binding and activation of platelets by VITT antibody/PF4 complexes.

Our results have contributed to a better understanding of pathogenesis in VITT which may lead to the development of disease biomarkers, improved diagnosis, new more efficacious therapies for VITT and by consequence, better clinical outcomes for the patients.

## Methods

### Human samples
VITT samples were collected from patients in Australia from the following hospitals: St George Hospital, Kogarah, Sydney, NSW; Calvary Mater Hospital, Wallsend, NSW; Box Hill Hospital, Box Hill, Victoria; University Hospital Geelong, Geelong, Victoria and Townsville University Hospital, Townsville, Queensland. Blood was collected from patients clinically diagnosed with HIT and VITT and positive for laboratory tests (ELISA and serotonin release assay)[2,39]. Healthy individuals vaccinated with ChAdOx1 nCoV-19 who did not develop VITT (Vax), patients with common venous thromboembolism (VTE), critically ill patients admitted to intensive care units (ICU) and healthy individuals were used as control groups. VITT patients were age (mean: 64 years, range: 46–84 years) and gender (4 males, 3 females) matched to healthy subjects vaccinated with ChAdOx1 nCoV-19 who did not develop VITT (mean: 62 years, range: 42–79 years) and patients with VTE (mean: 65 years, range: 48–77 years). Blood from healthy vaccinated group was collected 2–4 weeks of receiving ChAdOx1 nCoV-19 vaccine, matching the blood collection timeframe of VITT patient samples. For VTE patients, blood was collected during the acute phase. Diagnosis and treatment details of VITT, VTE and ICU groups are found in Supplementary Tables 1 and 2. This study was approved by the Human Research Ethics Committee of South Eastern Sydney Local Health District (17/211 LNR/17/POWH/501). Informed consent was obtained from all study participants. Sera and plasma samples were stored in aliquots at −80 °C until required for analysis.

### Diagnostic assays
The amount of anti-PF4 or anti-PF4/heparin antibodies in patient sera was measured using a solid phase PF4 (15 µg/mL) or PF4 (15 µg/mL)/heparin (0.1 U/mL) ELISA performed in microwell plates. Sera from patients or healthy individuals were added to each well and incubated for 60 min at room temperature and then washed. Conjugated anti-human IgG was added, incubated for 60 min at room temperature and washed. Chromogenic substrate reaction was stopped with 1 M $H_2SO_4$. Optical density was measured using an automatic plate reader (Tecan Infinite Pro).

$^{14}C$ serotonin-release assay ($^{14}C$-SRA) was performed as previously described[40]. Briefly, washed donor platelets were incubated with radiolabelled $^{14}C$ and heat inactivated patient's sera, in the presence or absence of PF4 (10 µg/mL), 0.1 U/mL heparin, IV.3 antibody (50 µg/mL) or 100 U/mL heparin, for 60 min at room temperature while stirring. Reaction was stopped using PBS-EDTA buffer and centrifuged. Radioactivity (counts per minute) of the supernatant was measured using a beta-counter. Levels >20% were considered positive.

## VITT IgG induces thrombosis in FcγRIIa⁺/hPF4⁺ mice

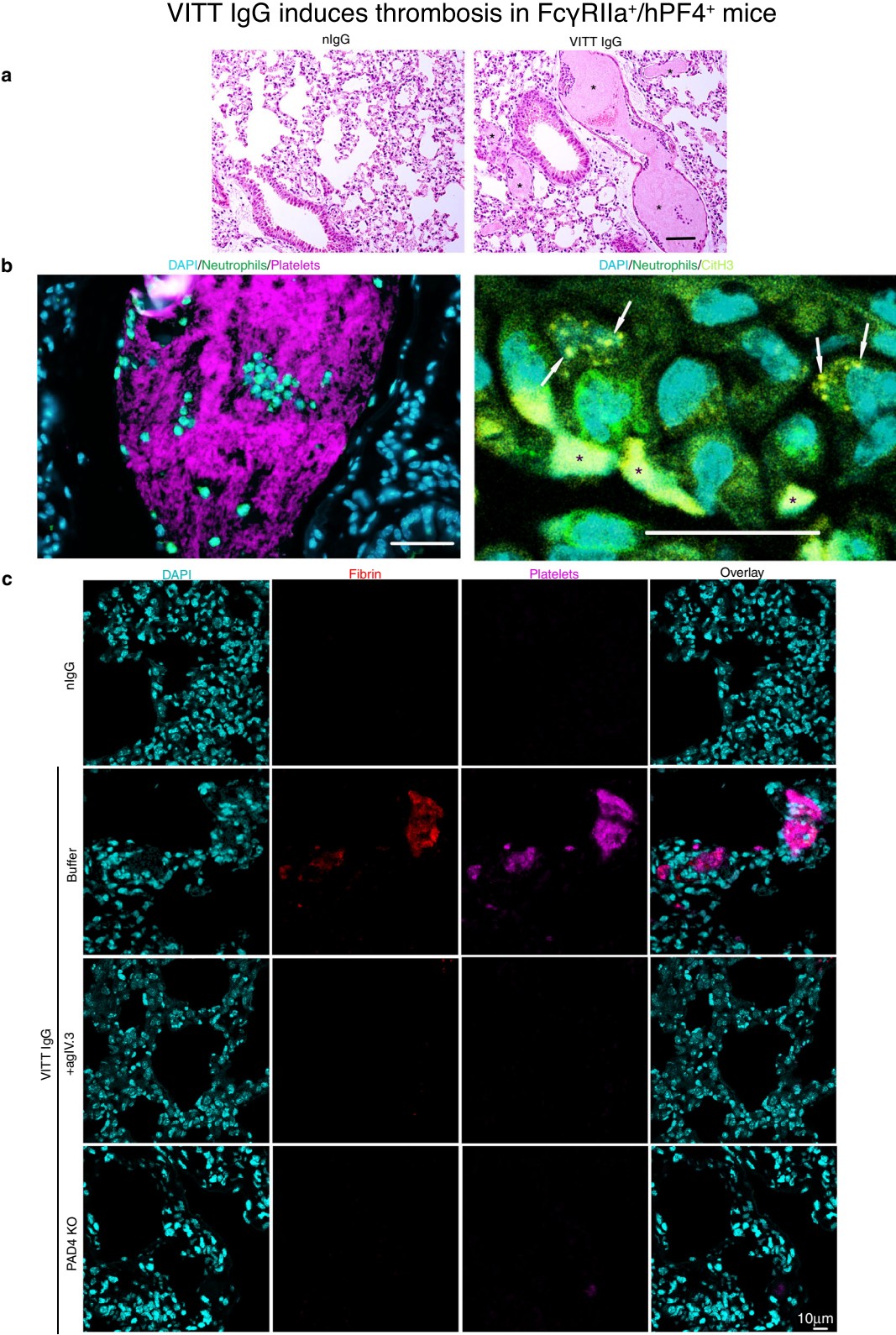

**Fig. 3 | VITT IgG induces thrombosis in FcγRIIa⁺/hPF4⁺ mice. a** Representative H&E staining of lung sections of mice treated with nIgG or VITT IgG. Scale bar 50 μm. **b** Fluorescent images of lung sections of mice treated with VITT IgG. Platelets and CitH3 were labelled in vivo with anti-CD42c-Dylight 649 (magenta) and AF594 (yellow). Neutrophil were stained with anti-Ly6G (green). Neutrophil infiltration in the clot is shown. Cell nuclei were stained with DAPI (blue). White arrows indicate areas of CitH3 staining. Asterisks indicate autofluorescence from red blood cells. Scale bars 50 μm. **c** Fluorescent images of lung sections of FcγRIIa⁺/hPF4⁺ mice treated with nIgG, VITT IgG or VITT IgG plus agIV.3 or FcγRIIa⁺/hPF4⁺/PAD4⁻/⁻ mice treated with VITT IgG. Fibrin labelled with AF594 (red) resulted from injection of AF594-labelled fibrinogen at 1 μg/g. Platelets were labelled in vivo with anti-CD42c-Dylight 649 (magenta). Cell nuclei were stained with DAPI (blue). Experiment was repeated independently (**a**) at least 10 times; **b**, **c** 3 times. Scale bar 10 μm. nIgG normal IgG, agIV.3 aglycosylated IV.3 antibody.

# Thrombosis and thrombocytopenia

**a**

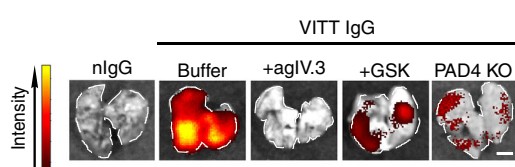

**b**

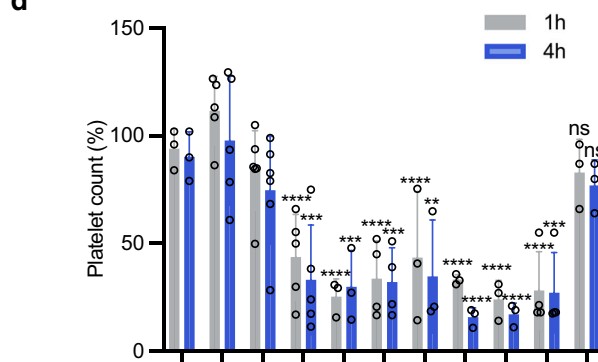

**c**

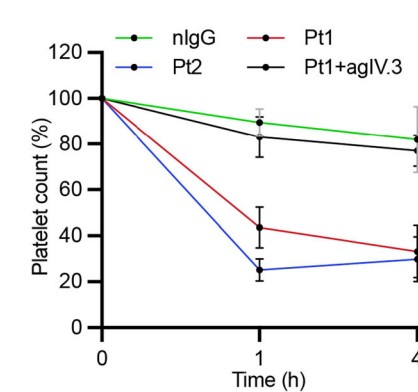

**d**

**e**

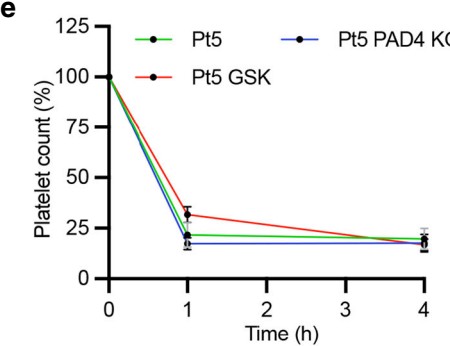

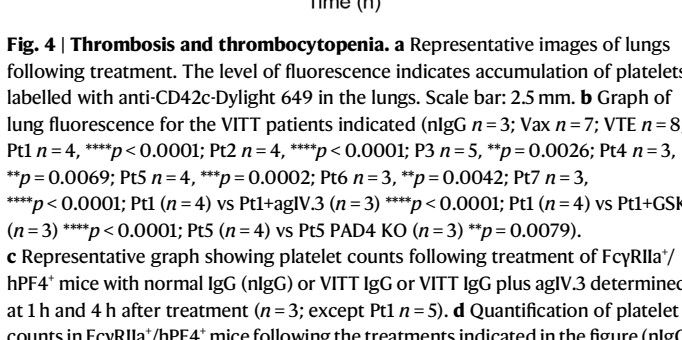

**Fig. 4 | Thrombosis and thrombocytopenia. a** Representative images of lungs following treatment. The level of fluorescence indicates accumulation of platelets labelled with anti-CD42c-Dylight 649 in the lungs. Scale bar: 2.5 mm. **b** Graph of lung fluorescence for the VITT patients indicated (nIgG $n = 3$; Vax $n = 7$; VTE $n = 8$; Pt1 $n = 4$, $****p < 0.0001$; Pt2 $n = 4$, $****p < 0.0001$; P3 $n = 5$, $**p = 0.0026$; Pt4 $n = 3$, $**p = 0.0069$; Pt5 $n = 4$, $***p = 0.0002$; Pt6 $n = 3$, $**p = 0.0042$; Pt7 $n = 3$, $****p < 0.0001$; Pt1 ($n = 4$) vs Pt1+agIV.3 ($n = 3$) $****p < 0.0001$; Pt1 ($n = 4$) vs Pt1+GSK ($n = 3$) $****p < 0.0001$; Pt5 ($n = 4$) vs Pt5 PAD4 KO ($n = 3$) $**p = 0.0079$). **c** Representative graph showing platelet counts following treatment of FcγRIIa⁺/hPF4⁺ mice with normal IgG (nIgG) or VITT IgG or VITT IgG plus agIV.3 determined at 1 h and 4 h after treatment ($n = 3$; except Pt1 $n = 5$). **d** Quantification of platelet counts in FcγRIIa⁺/hPF4⁺ mice following the treatments indicated in the figure (nIgG $n = 3$; Vax $n = 5$; VTE $n = 6$; Pt1 $n = 5$, $****p < 0.0001$, $***p = 0.003$; Pt2 $n = 3$,

$****p < 0.0001$, $***p = 0.001$; Pt3 $n = 4$, $****p < 0.0001$, $***p = 0.0005$; Pt4 $n = 3$, $****p < 0.0001$, $**p = 0.0023$; Pt5 $n = 3$, $****p < 0.0001$; Pt6 $n = 3$, $****p < 0.0001$; Pt7 $n = 4$ $n$, $****p < 0.0001$, $***p = 0.0002$; Pt1+agIV.3 $n = 3$, $ns$). **e** Graph showing platelet counts following treatment of FcγRIIa⁺/hPF4⁺ mice with VITT IgG with or without GSK or FcγRIIa⁺/hPF4⁺/PAD4⁻/⁻ mice plus VITT IgG determined at 1 h and 4 h after treatment (Pt5, Pt5 PAD4 KO $n = 3$; Pt5+GSK $n = 4$). Data presented as **b**, **d** mean ± SD; **c**, **e** mean ± SEM. Statistics. **b** One-way ANOVA with Dunnet's test for multiple comparisons. Unpaired two-tailed $t$-test for comparison between Pt5 in FcγRIIa⁺/hPF4⁺ and Pt5 in FcγRIIa⁺/hPF4⁺/PAD4⁻/⁻ mice. **d** One-way ANOVA with Dunnet's test for multiple comparisons. nIgG normal IgG, Vax healthy vaccinated subject IgG, VTE venous thromboembolism patient IgG, PAD4 KO PAD4 knockout FcγRIIa⁺/hPF4⁺ mice, agIV.3 aglycosylated IV.3 antibody, Pt patient. Source data are provided in the Source Data file.

## Antibodies

Purification of immunoglobulin G antibodies from patients' or healthy donor's sera was performed using Protein G Agarose (Roche Mannheim, Germany). The AKTA purifier chromatography system (GE Healthcare) was used for purification. Eluted peak fractions were pooled and concentrated using ultracentrifugal units. Purity of IgG was >95% as determined by SDS PAGE Gel analysis. Aggregation status of purified IgG was analysed by size exclusion chromatography using the NGC Chromatography System (Bio-Rad). Briefly, the columns were calibrated for molecular weights of proteins using the Column Performance Check Standard, Aqueous SEC 1 ladder (Phenomenex). VITT IgG sample was loaded into the column and the size of the eluted peaks were measured at absorbance 280 nm. We used heat-aggregated IgG as positive control. Functional activity of purified IgG was determined by platelet aggregation and serotonin release assays. Anti-PF4 IgG was purified from total patient IgG. PF4 purified from human platelets were incubated with biotinylation reagent then coupled to streptavidin magnetic beads (New England Biolabs). Purified total VITT IgG was incubated with streptavidin conjugated PF4 mixture for 90 min at 37 °C under gentle rotation. Using a magnetic separation rack, the undesired sample fraction was discarded. The retained PF4-specific VITT IgG was then eluted from the PF4-magnetic beads using acidic elution buffer (0.1 M glycine) followed by immediate neutralisation using 1 M Tris. Affinity purified anti PF4 IgG was quantitated by SDS gel electrophoresis. Hybridoma cells producing IV.3 were obtained from ATCC (clone HB-217). Cells were cultured in DMEM medium containing 10% FBS at 37 °C, 5% $CO_2$. Cells were cultured in serum-free DMEM 24 h prior to collection of antibody-containing supernatant. Protein G Sepharose affinity chromatography was used to purify IV.3.

## Platelet aggregation

Light transmission platelet aggregometry was used to determine antibody activity and role of FcγRIIa in VITT-induced platelet aggregation. Platelet-rich plasma (PRP) was prepared from citrate-anticoagulated healthy donor blood by centrifugation at room temperature at 150 $g$ for 10 min. 50 μL of VITT or control sera was added to a cuvette with 300 μL of PRP with or without FcγRIIa-inhibitor, IV.3 (20 μg/mL), whilst stirring at 37 °C for 15 min. Platelet poor plasma was used as blank.

## Quantification of NETosis markers

Cell-free DNA was measured in plasma of VITT and control samples using Quant-iT™ PicoGreen™ dsDNA assay kit (P11496, Invitrogen), as described by the manufacturer. Plasma levels of myeloperoxidase and citrullinated histone H3 were determined using the human myeloperoxidase ELISA kit (ab119605, Abcam) and H3R8Cit ELISA Capture and Detection kit (R&D143002, EpiCypher)[18], respectively, following the manufacturer's instructions.

## Cell isolation

Neutrophils and monocytes were purified using EDTA-anticoagulated blood and the EasySep Direct Human Neutrophil Isolation kit (19666, StemCell Technologies) or EasySep Human Monocyte Isolation kit (19359, StemCell Technologies) following the manufacturer's instructions. Purified neutrophils and monocytes were free of platelets and other blood cells as assessed by flow cytometry. Washed platelets were prepared from citrate-anticoagulated blood. For low density granulocytes, whole blood was diluted with PBS and Lymphoprep (07851, StemCell Technologies) was gently underlayed to avoid mixing with the diluted blood. Sample was then centrifuged at 800 × g for 20 min at room temperature. Peripheral blood mononuclear cell layer was harvested.

## Flow cytometry

Fresh citrate-anticoagulated blood from VITT patients or healthy donors was diluted with PBS. Platelet-neutrophil aggregates were analysed using anti-CD15 (1:50, Alexa Fluor 647, BD 562369) and anti-CD41a (1:10, PE, BD 555467), NETs were identified using anti-citrullinated histone H3 (1:50, ab5103), anti-MPO (1 μg/mL, PE, BD 341642) and goat anti-rabbit IgG (1:400, BV421, BD 565014). Monocytes and low density granulocytes were identified using anti-CD14 (1:50, V500, BD 561391) and anti-CD15 (1:50, Alexa Fluor 647, BD 562369) or anti-Ly6G (1:100, V450, BD560603) and anti-CD11b (1:100, PE, BD 557397). Platelet counts in mouse blood were determined by number of events acquired in 60 s relative to time 0. Flow cytometry data were analysed using FlowJo software (LCC, USA).

## Timelapse

Purified neutrophils were stained with Hoechst 33342 (5 μg/mL, 14533, Sigma) and seeded into eight-well Nunc Lab-Tek II chambers. Purified VITT IgG (5 mg/mL) or control IgG (5 mg/mL) with PF4 (12 μg/mL) were added to each reaction. Release of extracellular DNA was measured using Sytox Green (S7020, Invitrogen). Wells were imaged using a confocal laser-scanning microscope (Leica TCS SP8). Sytox green fluorescence relative to Hoechst 33342 fluorescence was calculated with ImageJ software (version 2.1.0/1.53c, NIH).

## Microfluidics

Citrate-anticoagulated blood was diluted 1:2 with PBS, supplemented with purified IgG (VITT IgG 3 mg/mL, control IgG 3 mg/mL) and incubated at 37 °C for 90 min. In selected experiments, blood was pre-incubated with IV.3 (20 μg/mL) or DNase I (160 U/mL). Blood was stained with combinations of Hoechst 33342 (3 μg/ml), Sytox green (0.3 μM), anti-CD41 Alexa 647 (15 μg/mL), anti-CD41-FITC (15 μg/mL), anti-CD15 Alexa 647 (15 μg/mL), anti-fibrin Alexa 594 (30 μg/mL), anti-CitH3 Alexa 594 (30 μg/mL) prior to perfusion through Vena8 Fluoro + ™ biochip microchannels coated with vWf (Haematologic Technologies United BioResearch Products Pty Ltd). Biochips were mounted on a fluorescent microscope (Zeiss Axio Observer.A1) and fluorescence images from different microscopic fields were captured in real time with a Q-Imaging EXi Blue™ camera (Q-Imaging, Surry, BC, Canada) with the fluid shear stress set at 67 dyne/cm² (shear rate 1500/s) for 30 min. Selected samples were fixed with 2% paraformaldehyde and imaged by confocal microscopy.

## Mouse model

Mice were housed under a 12 h light/12 h dark cycle at 21 °C with 50% humidity. Mice expressing the $R^{131}$ isoform of human FcγRIIa and human PF4 were generated in C57BL/6 background. Double transgenic (FcγRIIa⁺/hPF4⁺) and FcγRIIa⁺/hPF4⁺/PAD4⁻/⁻ have been characterised previously[11,41]. VITT was recreated in these mice by intravenous injection of purified VITT IgG (250 μg/g). Inhibitors of NETosis (GSK484 2μg/g, Cayman chemicals) or anti FcγRIIa (aglycosylated IV.3, 1 μg/g) were injected at time 0. Anti-CD42c Dylight-649 antibody (1 μg/g, Emfret, Germany) and Alexa Fluor 594-fibrinogen (4 μg/g) were used to label mouse platelets and fibrin in vivo, respectively. Anti-Ly6G Alexa 488 (0.5 μg/g, BioLegend) and anti-CitH3 Alexa 594 (0.3 μg/g) were injected intraperitoneally 45 min after VITT IgG administration for select experiments for in vivo labelling of mouse neutrophils and CitH3. Following euthanasia, lungs were perfused with PBS followed by formalin, extracted and imaged using the IVIS Spectrum (Perkin Elmer). Fluorescence was calculated in radiant efficiency using Living Image 4.5.5 software (Perkin Elmer). All animal experiments were approved by the University of New South Wales Animal Care and Ethics Committee.

## Histology

Formalin-fixed lungs were embedded in paraffin, sectioned at 4 microns and mounted onto slides. Slides were deparaffinised, rehydrated, and underwent heat-induced antigen retrieval. Slides were

probed with anti-Ly6G (1:100, Alexa Fluor 488, 127626 BioLegend). Vectashield antifade mounting medium with DAPI (H-1200, Vector Laboratories) was used to mount glass coverslips onto the slides. Slides were imaged by confocal microscopy. Slides were also stained with H&E and imaged with a Zeiss Axioskop microscope.

## Statistical analyses

Statistical tests were performed using GraphPad Prism version 8 (GraphPad Software, USA). The following statistical tests were used in this study as described in the figure legends: (1) Shapiro-Wilk normality test. (2) Student's *t*-test was performed when comparing between two groups. (3) Multiple comparisons were analysed by one-way ANOVA with post-test correction for multiple comparisons. Each individual healthy donor for in vitro experiments and each mouse used for animal experiments was considered a biological replicate. *P*-values < 0.05 were considered statistically significant.

## Reporting summary

Further information on research design is available in the Nature Research Reporting Summary linked to this article.

## Data availability

Data generated from this study are available from the corresponding author upon reasonable request. Source data underlying all relevant figures are provided with this paper. Source data are provided with this paper.

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

## Acknowledgements
The authors wish to thank Steven McKenzie (Philadelphia, USA) for providing FcγRIIa⁺/hPF4⁺ mice, Drs Feng Yan, Rose Wong and Kathryn Evans for valuable technical assistance, Drs Sumita Ratnasingam, John Casey, and Jay Hocking for management of VITT patients and valuable clinical input, O Szeto, J Bennett, M Poxton, E Heyer and P Rojanski for assistance in obtaining human research ethics/governance approvals, and members of the THANZ VITT Advisory Group for helpful discussion of VITT patients. The authors acknowledge the facilities and technical assistance of the National Imaging Facility, a National Collaborative Research Infrastructure Strategy (NCRIS) capability, at the Biological Resources Imaging Laboratory, University of New South Wales. This work was supported by grants from National Health and Medical Research Council, Australia, Program Grant APP1052616 and New South Wales Capacity Program Senior Researcher Grant RG201677 to B.H.C.; NSW Health Cardiovascular Disease Clinician Scientist Grant and National Health and Medical Research Council Australia, Investigator Grant to J.J.H.C.

## Author contributions
B.H.C. conceived the idea, designed and supervised the research, analysed the data and wrote the manuscript, H.H.L.L. and J.P. designed and carried out the experiments, collected and analysed the data and wrote the manuscript, Z.A. performed platelet function assays and microfluidic studies, collected and analysed the data, F.N.R. carried out histology and immunochemistry studies, collected and analysed the data, J.J.H.C. provided conceptual input, designed experiments and analysed data, S.S.Z., S.B.T. and A.E. provide intellectual input, analysed clinical data and managed VITT patients. All authors reviewed and edited the manuscript and approved the final version of the manuscript.

## Competing interests
The authors declare no competing interests.
