## [Peer Review File · Nature Communications]

Reviewers' Comments:

Reviewer #1:

Remarks to the Author:

This is an interesting manuscript that attempts to show that NET formation is the major reason for the development of thrombosis and thrombocytopenia in a subset of patients, who develop these conditions after COVID-19 vaccination. The authors analyzed the relevant blood samples of 7 patients, and also conducted in vitro and in vivo mouse model experiments. Although data appear to support the hypothesis, there are number of points need to be verified.

1. Fig. 1- Independently measuring CitH3 and cfDNA does not confirm the presence of NETs because macrophage or monocytes and other cells could also generate CitH3 (Fig. 1d and e). Hence a sandwich ELISA typical for NETs are needed to confirm this point. Although CD15+ granulocytes with MPO represent neutrophils (Fig.1i), MPO-CitH3 double staining is reported in macrophages or monocytes undergoing ET formation. Hence, other sources of ETs from various cells needs to be confirmed. NPA and NETs are shown only for 3 of the 7 patients (Figs. 1h,j). The reasons for not including all the patients in these assays require clarification.
2. Fig. 2- Figs. 2a,b – how were the NETs identified? Why LDG and NETs were not analyzed together? Fig. 2d- again, why not all the patient samples were not used in these assays (only 4 of 7 are shown). Is it a true thrombi (as stated in the manuscript) or a platelet/neutrophil aggregate, because the blood used in the in vitro system should have had anti-coagulants (citrate) or components deficient in clotting factors (sera) (Figs.2e,f); furthermore, although a large number of neutrophils were seen, only a small amounts of CitH3 is visible. Hence, the involvement of NET formation in this process in not certain. It is unclear why the analyses was done only on DNA, platelet and neutrophils but not on CitH3; DNA-platelet-neutrophil may not represent NETs because the CitH3 was present only in some area (Figs. 2e,f).
3. Fig. 3 - Fig.3a – lung tissue locations used for the comparison of nIgG vs VITT IgG conditions appear uncertain. The location for nIgG looks like the lower airways with normal alveoli; however, the location used for illustrating the effect of VITT IgG looks very different. Furthermore, blood normally does not leak into the alveoli; Also, the clots seem to have only a few neutrophils, and the NET formation under this condition is not discernable from this image. The images imply that neutrophils are trapped within in a huge platelet aggregate. Quantitative analyses to inform the frequency of these structures in the lungs are needed to objectively interpret the data.
4. Fig. 4 – Fig 4a shows strong platelet staining under GSK(+) and PAD4(-) conditions, albeit low; however, no platelet was visible in Fig. 3c. What is the reason for this discrepancy?
5. Fig. E1. What is the location of platelets in panel c?
6. Fig. E2. PAD4 is needed for NADPH oxidase independent NET formation. Hence, the involvement of this pathway needs to be verified in the manuscript, by experimentation. Also, NET release was shown independent of the neutrophil nuclear DNA. Do the authors imply mitochondrial DNA release (e.g., vital NETosis) (panel c)?

Reviewer #2:

Remarks to the Author:

The authors propose a mechanism for vaccine-induced thrombosis and thrombocytopenia occurring after SARS-CoV-2 vaccination with adenoviral vectors. The demonstrate that neutrophil extracellular traps (NETs) are involved in the pathogenesis of this condition. The following comments may be made:

1. VITT patients have a remarkably high rate of cerebral sinus vein thrombosis, which remains unexplained. It would be interesting if the authors could indicate whether the occurrence of (vascular bed specific) NETs could be a factor in this peculiar manifestation.
2. It is not clear why the authors think that thrombocytopenia in VITT is caused through NET-induced platelet activation, when the target molecule of the auto-antibody response is platelet factor-4, which is abundantly present in platelets. A much more straightforward explanation could be that the antibodies directly cause platelet activation. They should add experiments showing that this is the case or not.

3. The findings in 7 affected patients were contrasted with measurements in apparently unmatched healthy volunteers. I believe this is not the right control group. Better controls might be patients with common venous thromboembolism, other forms of critical illness or vaccinated subjects without VITT. I would strongly urge the authors to include such control groups to make their findings more robust.

4. It should be emphasized that in contrast to several statements in the text, the authors did not directly demonstrate the presence of NETs in their subjects. They have measure plasma levels of NET constituents, which is a surrogate marker (with not precisely known accuracy) for the presence of NETs.

5. The timing of blood sampling in patients with VITT is unclear. Were bloodsamples collected at admission or later on during the clinical course? Had patients received any treatment, such as steroids or immunoglobulins

Reviewer #3:

Remarks to the Author:

The group of Dr. Chong presents a comprehensive study on the mechanisms of platelet activation and thrombus formation in VITT. The manuscript provides experimental evidence that total IgG obtained from VITT patients induces thrombosis and thrombocytopenia in a humanized (FcγRIIa+/hPF4+) murine model in vivo. While the study has been systematically performed, the key results presented in the paper on the involvement of platelet FcγRIIa and NETosis are not novel per se but very important confirmation of recently published detailed in vitro studies and mechanistic insights of VITT.

Immune complexes, innate immunity, and NETosis in ChAdOx1 vaccine-induced thrombocytopenia.

Holm S, Kared H, Michelsen AE, Kong XY, Dahl TB, Schultz NH, Nyman TA, Fladeby C, Seljeflot I, Ueland T, Stensland M, Mjaaland S, Goll GL, Nissen-Meyer LS, Aukrust P, Skagen K, Gregersen I, Skjelland M, Holme PA, Munthe LA, Halvorsen B. Eur Heart J. 2021 Aug 18:ehab506. doi: 10.1093/eurheartj/ehab506. Online ahead of print.

and on the mechanism of VITT reported in April as preprint and recently published in BLOOD. <https://assets.researchsquare.com/files/rs-440461/v1/0f1d7cd8-a707-49db-a136-3ebc62f65467.pdf?c=1631881427>

Insights in ChAdOx1 nCov-19 Vaccine-induced Immune Thrombotic Thrombocytopenia (VITT). Greinacher A, Selleng K, Palankar R, Wesche J, Handtke S, Wolff M, Aurich K, Lalk M, Methling K, Volker U, Hentschker C, Michalik S, Steil L, Reder A, Schönborn L, Beer M, Franzke K, Büttner A, Fehse B, Stavrou EX, Rangaswamy C, Mailer RK, Englert H, Frye M, Thiele T, Kochanek S, Krutzke L, Siegerist F, Endlich N, Warkentin TE, Renné T. Blood. 2021 Sep 29:blood.2021013231. doi: 10.1182/blood.2021013231

The present study extends these findings and the understanding of the mechanism of VITT by the first mouse model.

The authors may consider to cite the work of others as confirmation of the same concept by two or three independent groups is highly valuable. Furthermore, the wording in the sections "main" and in the discussion should be adjusted accordingly.

Minor comments:

1. According to the authors, total IgG from VITT patients (termed VITT IgG) was purified by Protein G affinity chromatography, and SDS-PAGE checked its purity. While this is standard practice for obtaining total IgG, no attempts were made to assess the aggregation status of the purified total IgG by native gel electrophoresis. This is relevant to exclude the potential artifacts of aggregated IgG in all experiments (both ex vivo and in vivo) involving platelets in PRP or whole blood experiments and including those with neutrophils and murine models with FcγRIIa+/hPF4+.

2. Experiments related to VITT IgG induced thrombosis in vivo:

It is unclear from the manuscript why an extremely high concentration of the purified total IgG from VITT patients was injected into the murine model in vivo (250 µg/g of the mouse).

Intriguingly in the same experiments, inhibition of FcγRIIa with deglycosylated IV.3 in vivo is performed with 1 µg/g.

In addition, several ex vivo and in vitro experiments use a very high concentration of purified total IgG from 3mg/mL up to 5mg/mL, and inhibitions with IV.3 work at much lower concentrations. It is difficult to conclude the biological relevance of using very concentrations of total IgG from VITT patients in both ex vivo and in vivo experiments described in the manuscript. Critically, no attempts have been made to quantify specific anti-PF4 IgG concentrations or their typing in the patient sera, or the protein G purified total IgG.

3. Microfluidic whole blood experiments and in vivo murine model: Did the total IgG isolated from VITT patient sera produce NETosis under shear in the absence of added PF4? What is the endogenous concentration of PF4 FcγRIIa+/hPF4+ murine model under healthy conditions (i.e., non-inflammation)? Is it comparable to a healthy individual since the total IgG purified from VITT patients appears to induce platelet aggregation in PRP and NETosis in the absence of added PF4 throughout the experiments presented in the manuscript?

4. General comment: It is kindly advised to report the detailed protocol of each experimental method as per journal policy and for replication studies.

E.g., GSK484 inhibition: Please specify the concentration used across different experiments, PF4 and PF4/H EIA studies, imaging studies.

5. Discussion: anti-PF4 antibodies are frequent and it is a decade old concept in HIT that these antibodies which usually do not activate platelets are clinically irrelevant. For the same reason the presence of anti-PF4 antibodies in individuals without thrombosis had not created doubts about the concept of VITT, at least to the knowledge of this reviewer. The authos may consider to revise this part of the discussion.

REVIEWER COMMENTS

Reviewer #1 (Remarks to the Author):

1) This is an interesting manuscript that attempts to show that NET formation is the major reason for the development of thrombosis and thrombocytopenia in a subset of patients, who develop these conditions after COVID-19 vaccination. The authors analyzed the relevant blood samples of 7 patients, and also conducted in vitro and in vivo mouse model experiments. Although data appear to support the hypothesis, there are number of points need to be verified.

1.1 Fig. 1- Independently measuring CitH3 and cfDNA does not confirm the presence of NETs because macrophage or monocytes and other cells could also generate CitH3 (Fig. 1d and e). Hence a sandwich ELISA typical for NETs are needed to confirm this point. Although CD15+ granulocytes with MPO represent neutrophils (Fig. 1i), MPO-CitH3 double staining is reported in macrophages or monocytes undergoing ET formation. Hence, other sources of ETs from various cells needs to be confirmed.

We agree with Reviewer 1 that measuring CitH3, cfDNA, MPO and even MPO-CitH3 double staining does not necessarily confirm the presence of NETs as these materials are not exclusively released by activated neutrophils during NETosis but they are also produced by monocyte/macrophage when they undergo ETosis. Other ET-producing cells, due to their low abundance in blood, presumably contribute insignificant amounts of ETs. However, cell-appendant MPO-CitH3 double staining in Fig. 1k were measured by gating neutrophils using flow cytometry. Therefore, these ETs components are those released specifically by neutrophils and not by monocytes/macrophages. We are justified to conclude that detection of these neutrophil-specific MPO-CitH3 suggests the presence of NETs.

There are no commercial ELISA that detect the cellular source of ETs^{1,2}, whether from neutrophils, monocytes or other cells. As suggested by the reviewer, we have used commercial CitH3 and MPO sandwich ELISAs typical for NETs (Fig. 1d,e). These have been validated to be accurate and reliable and used to measure NETs markers in this and related fields of research³⁻⁵. As neutrophils are much more numerous in the blood compared to monocytes, the total plasma ETs measured by sandwich ELISA are likely to be predominantly NETs.

The authors recognise that monocytes may express CD15 at low levels^{6,7} and neutrophils can express CD14 in diseased states⁸. Supplementary Fig. 1a (purified neutrophils), b (purified monocytes), and c (overlay) confirm that the neutrophil and monocyte populations are positioned in distinct FSC/SSC position and have specific CD15, CD14 and CD66b expression⁹. The gating strategy demonstrates that neutrophils can be clearly distinguished from monocytes by their FSC/SSC profile and CD15 expression. We have performed additional experiments to further clarify the confusion between neutrophils and monocytes and the ETs they produce.

As suggested by the reviewer, we analysed different sources of ETs. Using purified cells, we found that neutrophils (Figure 2c-d) and monocytes (Suppl. Fig. 1e) produced MPO-CitH3 suggesting both cell types release ETs following activation by VITT immune complexes.

1.2 NPA and NETs are shown only for 3 of the 7 patients (Figs. 1h,j). The reasons for not including all the patients in these assays require clarification.

Figures 1j and l (previously Figs. 1h,j) represent NPA and NETs present in fresh VITT patients' blood. These ex vivo flow cytometry assays require analysis of fresh blood within 3 hours of collection. VITT cases in this study were recruited from hospitals in different cities in Australia, a vast country. Due to long distances between some participating hospitals and the host hospital, ex vivo assays using fresh blood could not be conducted on all 7 patients. Unfortunately, this difficulty was exacerbated by the strict COVID-19 lockdown and closure of Australian state borders. As a result, analysis on fresh blood could not be conducted on 4 patients. Since the number of recipients of AstraZeneca vaccine has dropped substantially in Australia, cases of VITT are now rare and recruitment of new cases is now not feasible.

2.1 Figs. 2a,b – how were the NETs identified? Why LDG and NETs were not analyzed together?

Neutrophils (CD15+) positive for both MPO and CitH3 were defined as NETting neutrophils¹⁰⁻¹².

LDG and NETs are analysed together using flow cytometry which detected NETs (co-expression of CitH3 and MPO) in LDG cell population. Figure 1m has been updated to include these data. It is unnecessary to show this again in Figure 2. Few LDG were present in controls so data may not be reliable from so few LDG.

2.2 Fig. 2d- why not all the patient samples were not used in these assays (only 4 of 7 are shown).

The authors have updated Figure 2 to include data for all 7 patients.

2.3 Is it a true thrombi (as stated in the manuscript) or a platelet/neutrophil aggregate, because the blood used in the in vitro system should have had anti-coagulants (citrate) or components deficient in clotting factors (sera) (Figs.2e,f); furthermore, although a large number of neutrophils were seen, only a small amounts of CitH3 is visible. Hence, the involvement of NET formation in this process is not certain. It is unclear why the analyses was done only on DNA, platelet and neutrophils but not on CitH3; DNA-platelet-neutrophil may not represent NETs because the CitH3 was present only in some area (Figs. 2e,f). In the experiments in Figs. 2e-f, the blood was collected in tubes containing citrate and citrated blood did not prevent platelet¹³ and white cell¹¹ activation and thrombus formation^{14,15}.

The structures shown in Figs. 2e-f contained neutrophils, platelets, DNA and fibrin. Platelet aggregates and fibrin are two essential components of a thrombus by any conventional definitions. The presence of neutrophils and extracellular DNA in thrombi has been increasingly reported¹⁶⁻¹⁸. These structures had the characteristics of true thrombi, and were not just platelet/neutrophil aggregates.

We believe that CitH3 was present in amounts consistent with (and more prominent than other) observations in the literature¹⁹⁻²⁷ to conclude NETs involvement in VITT-induced thrombosis. CitH3 is dispersed during blood flow through the microfluidics chamber and not locally concentrated, and it is not as prominent as aggregated neutrophils, platelets and fibrin that adhered to the microchamber wall. Due to the dispersed nature of CitH3, it was not

included in area coverage analysis. As only a proportion of neutrophils underwent NETosis in the flowing blood at a given time when the image was taken in this dynamic system, it is expected that the presence of CitH3 would be less than the total neutrophils present.

3. Fig.3a – lung tissue locations used for the comparison of nIgG vs VITT IgG conditions appear uncertain. The location for nIgG looks like the lower airways with normal alveoli; however, the location used for illustrating the effect of VITT IgG looks very different. Furthermore, blood normally does not leak into the alveoli; Also, the clots seem to have only a few neutrophils, and the NET formation under this condition is not discernable from this image. The images imply that neutrophils are trapped within in a huge platelet aggregate. Quantitative analyses to inform the frequency of these structures in the lungs are needed to objectively interpret the data.

The authors acknowledge that H&E images used in Fig. 3a show different locations of the lung. We have used new images of comparable lung areas. We show thrombi located in blood vessels in the lung, not in the alveoli. The authors agree that the frequency of thrombosis in VITT and controls is needed and is now included in Supplementary Figure 4a.

Figure 3 histology of lung sections show the presence of neutrophils within clots in sufficient numbers to suggest they are involved in the clotting process and the neutrophil numbers were comparable with those in previous published reports on the same topic^{4,16,18,28}. As in previous reports, the neutrophils were present in clusters and this suggest cell activation and cell-cell interaction as expected during thrombosis. If they were passively trapped they would be expected to be more spread out. The number of neutrophils present in the clots varies from clot to clot and from the types of clots; there is no known number of neutrophils that must be present to favour active participation versus being passively trapped. Histology tissue section examination is one piece of data to support the overall collection of robust data we presented to strongly support the role of activated neutrophils and NETosis in VITT-antibody induced thrombosis

To confirm NETs formation, lung sections were stained with anti-Ly6G (neutrophils) and anti-CitH3 and imaged by confocal microscopy. This is now added to Figure 3b and confirms the presence of CitH3 in neutrophils in the lungs of VITT IgG-treated mice.

4. Fig 4a shows strong platelet staining under GSK(+) and PAD4(-) conditions, albeit low; however, no platelet was visible in Fig. 3c. What is the reason for this discrepancy?

Fig. 4a shows that platelets are still present within the lung of GSK-treated and PAD4 KO mice but at very low levels (high accumulation is represented in yellow, very low in burgundy), as shown in Fig. 4b (quantification of Fig. 4a). This is supported by the very faint small platelet aggregates present in the lung section of PAD4 KO mice in Figure 3c. The apparent discrepancy is probably due to the different techniques/approaches used to capture the signals. Confocal microscopy of lung sections (Fig. 3c) and whole organ CT scan imaging (Fig. 4a) are two imaging techniques/approaches with different detection sensitivities, used for different purposes. Whole organ imaging detects overall fluorescence and is quantitative, while confocal images are illustrative/semi-quantitative and only represent a small area (less than 100 μm x 5 μm) of the lung. These two techniques have been used to complement each other.

This discrepancy does not substantially affect the strength/validity of our findings (as shown in Fig. 4b) that blocking PAD4 activity/NETosis significantly reduces thrombosis in VITT.

5. Fig. E1. What is the location of platelets in panel c?

Platelets (shown in purple, white arrows) are accumulated within the blood vessels throughout the lung of VITT IgG-treated mice. Suppl. Fig. 3c (previously Fig. E.1) illustrates the contrast between the presence of thrombosis in mice treated with VITT IgG compared and the control mice.

6. Fig. E2. PAD4 is needed for NADPH oxidase independent NET formation. Hence, the involvement of this pathway needs to be verified in the manuscript, by experimentation. Also, NET release was shown independent of the neutrophil nuclear DNA. Do the authors imply mitochondrial DNA release (e.g., vital NETosis) (panel c)?

Our current study has already included substantial amount of novel data regarding NETs and mechanisms of thrombosis in VITT and our study aims do not include investigation of NADPH oxidase and the issues of neutrophil nuclear versus mitochondrial DNA release. It would not be expected for a single paper to cover all possible aspects of VITT thrombosis mechanisms.

Although it would be interesting to determine the involvement of NADPH oxidase pathway in VITT-induced NETs, this is beyond the scope of the current study. We expect that in VITT, like in HIT, NET formation is dependent on both PAD4 and NADPH oxidase²⁹. The presence/involvement of mitochondrial DNA in VITT-induced NETosis is not suggested by our study and would also need to be confirmed in future studies.

Reviewer #2 (Remarks to the Author):

The authors propose a mechanism for vaccine-induced thrombosis and thrombocytopenia occurring after SARS-CoV-2 vaccination with adenoviral vectors. They demonstrate that neutrophil extracellular traps (NETs) are involved in the pathogenesis of this condition. The following comments may be made:

1. VITT patients have a remarkably high rate of cerebral sinus vein thrombosis, which remains unexplained. It would be interesting if the authors could indicate whether the occurrence of (vascular bed specific) NETs could be a factor in this peculiar manifestation. The authors expect that the VITT antibody activate cerebral vascular endothelial cells. We agree that it would be interesting to investigate this further, however, we are not currently in the position to perform the required experiments which are outside the scope of the current study. We hope to study this in future studies in collaboration with our colleagues with expertise and resources in endothelial cell research.

2. It is not clear why the authors think that thrombocytopenia in VITT is caused through NET-induced platelet activation, when the target molecule of the auto-antibody response is platelet factor-4, which is abundantly present in platelets. A much more straightforward explanation could be that the antibodies directly cause platelet activation. They should add experiments showing that this is the case or not.

The authors do not think and have not indicated that thrombocytopenia in VITT is caused

through NET-induced platelet activation. We agree with the reviewer that thrombocytopenia is due to platelet activation by VITT antibodies/PF4 complexes, shown by serotonin release assays and platelet aggregation assay, in the absence of neutrophils or other cells (Fig. 1b,c). In vivo, thrombocytopenia also occurs in the absence of NETosis (Fig. 4e). We have added the following sentence to the manuscript to highlight this point and included Suppl. Fig. 5b to illustrate that thrombocytopenia is caused by platelet activation.

“Thrombocytopenia is likely due to direct binding and activation of platelets by VITT antibody/PF4 complexes.”

3. The findings in 7 affected patients were contrasted with measurements in apparently unmatched healthy volunteers. I believe this is not the right control group. Better controls might be patients with common venous thromboembolism, other forms of critical illness or vaccinated subjects without VITT. I would strongly urge the authors to include such control groups to make their findings more robust.

The authors agree that vaccinated subjects without VITT would serve as more appropriate controls. We have now used vaccinated subjects without VITT as the control group for all experiments. We have also included common venous thromboembolism (VTE) as a comparative control group in all figures. A third control group (critically ill patients in intensive care) has also been added to relevant experiments (Figure 1f-m).

The age between VITT and control groups were matched.

Gender spread was matched for VITT, VTE and vaccine groups, 3 females and 4 males; ICU group had 5 females and 1 male (difficulty getting consent from gender matched patients for this group). All VTE and VITT patients received anticoagulant treatment. These data are now included in Supplementary Table 1 and 2.

4. It should be emphasized that in contrast to several statements in the text, the authors did not directly demonstrate the presence of NETs in their subjects. They have measure plasma levels of NET constituents, which is a surrogate marker (with not precisely known accuracy) for the presence of NETs.

The authors agree that measuring plasma levels of NETs markers is an indirect method. We would like to highlight our data directly demonstrating the presence of NETs in our VTTT patients.

Ex vivo evidence from VITT patients: In addition to plasma data, we did directly demonstrate the presence of NETs in fresh blood from VITT patients (Fig. 1k,l). Fresh VITT patient blood samples were analysed (without any treatment) by flow cytometry and they showed the presence of NETting neutrophils. These ex vivo data directly confirmed that neutrophils in VITT patient blood were undergoing NETosis. As stated in the response to Reviewer 1, it was not possible to analyse fresh blood from all 7 patients due to their locations in different Australian states (please also read our response to (2) reviewer 1's comments).

Ex vivo evidence from mice: Similarly, direct analysis of fresh blood from mice administered with VITT antibodies showed the presence of activated (low density) neutrophils (Suppl. Fig. 4b).

In vivo evidence from mice: Lung sections show the presence of CitH3 in mouse neutrophils (Fig. 3b).

In VITT patients, other potential sources of NETs include co-existing cell activation by putative immune complexes formed by viral spike proteins/antibodies and even vaccine-derived adenoviral particles as suggested by many commentators^{27,30}. However, in our study, we clearly show in the murine VITT model and VITT patient plasma that VITT Ab/immune complexes are the direct cause of NETosis and the subsequent thrombosis, findings not previously reported.

5. The timing of blood sampling in patients with VITT is unclear. Were blood samples collected at admission or later on during the clinical course? Had patients received any treatment, such as steroids or immunoglobulins

For all patients, samples were collected at or soon after admission pre-treatment. Additional blood was collected during course of treatment: blood was sampled within 2 weeks of admission for 6 patients, and within 3 weeks of admission for 1 patient. VITT patients received a variety of treatments: all 7 received anticoagulants (fondaparinux, argatroban, apixaban, dabigatran, bivalirudin, fondaparinux, warfarin), 5 received IVIg, 1 received dexamethasone, 1 received methylprednisolone (see Supplementary Table 1). Timing of blood sampling and treatment of VITT patients have now been added to the manuscript.

Reviewer #3 (Remarks to the Author):

The group of Dr. Chong presents a comprehensive study on the mechanisms of platelet activation and thrombus formation in VITT. The manuscript provides experimental evidence that total IgG obtained from VITT patients induces thrombosis and thrombocytopenia in a humanized (FcγRIIa+/hPF4+) murine model in vivo. While the study has been systematically performed, the key results presented in the paper on the involvement of platelet FcγRIIa and NETosis are not novel per se but very important confirmation of recently published detailed in vitro studies and mechanistic insights of VITT.

Immune complexes, innate immunity, and NETosis in ChAdOx1 vaccine-induced thrombocytopenia.

Holm S, Kared H, Michelsen AE, Kong XY, Dahl TB, Schultz NH, Nyman TA, Fladeby C, Seljeflot I, Ueland T, Stensland M, Mjaaland S, Goll GL, Nissen-Meyer LS, Aukrust P,

Skagen K, Gregersen I, Skjelland M, Holme PA, Munthe LA, Halvorsen B. Eur Heart J. 2021 Aug 18;ehab506. Doi: 10.1093/eurheartj/ehab506. Online ahead of print. and on the mechanism of VITT reported in April as preprint and recently published in BLOOD.

<https://assets.researchsquare.com/files/rs-440461/v1/0f1d7cd8-a707-49db-a136-3ebc62f65467.pdf?c=1631881427>

Insights in ChAdOx1 nCov-19 Vaccine-induced Immune Thrombotic Thrombocytopenia (VITT). Greinacher A, Selleng K, Palankar R, Wesche J, Handtke S, Wolff M, Aurich K, Lalk M, Methling K, Volker U, Hentschker C, Michalik S, Steil L, Reder A, Schönborn L, Beer M, Franzke K, Büttner A, Fehse B, Stavrou EX, Rangaswamy C, Mailer RK, Englert H, Frye M, Thiele T, Kochanek S, Krutzke L, Siegerist F, Endlich N, Warkentin TE, Renné T. Blood. 2021 Sep 29;blood.2021013231. doi: 10.1182/blood.2021013231

The present study extends these findings and the understanding of the mechanism of VITT by the first mouse model.

The authors may consider to cite the work of others as confirmation of the same concept by two or three independent groups is highly valuable. Furthermore, the wording in the sections “main” and in the discussion should be adjusted accordingly.

We agree with the reviewer and have cited the work of Holm *et al* and Greinacher *et al* in our original manuscript and again in the revised manuscript. We acknowledge the work of independent groups which provided evidence of NETs in VITT and have changed our wording accordingly in the main and discussion sections.

Minor comments:

1. According to the authors, total IgG from VITT patients (termed VITT IgG) was purified by Protein G affinity chromatography, and SDS-PAGE checked its purity. While this is standard practice for obtaining total IgG, no attempts were made to assess the aggregation status of the purified total IgG by native gel electrophoresis. This is relevant to exclude the potential artifacts of aggregated IgG in all experiments (both *ex vivo* and *in vivo*) involving platelets in PRP or whole blood experiments and including those with neutrophils and murine models with FcγRIIIa+/hPF4+.

The authors agree that the aggregation status of the purified total IgG should be determined to exclude potential artifacts. We analysed the purified IgG samples by size exclusion chromatography using the NGC Chromatography System (Bio-Rad, California). Briefly, the columns were calibrated for molecular weights of proteins using the Column Performance Check Standard, Aqueous SEC 1 ladder (Phenomenex, California). VITT IgG samples were loaded into the column and the size of the eluted peaks were measured at absorbance 280nm. We used heat-aggregated IgG as positive control. In heat-aggregated IgG 49.9% represented aggregates (MW > 600 kDa) and 50.1% of IgG existed as monomer. In contrast, in purified VITT and control IgG, we found that the 95.7% (range 92.2-98.1%) of purified IgG was present as monomers, with only 5.4% (range 1.9-7.5%) present as aggregates (MW ~ 300 kDa). The tiny amount (~5%) of small IgG aggregates present in normal total IgG did not cause any evidence of platelet or neutrophil activation in our > 30 years' experience in this and related fields of research. It is unavoidable to have a small percentage of aggregated IgG due to their tendency to aggregate. Chromatography traces are shown in Supplementary Fig. 2a.

2. Experiments related to VITT IgG induced thrombosis in vivo:

It is unclear from the manuscript why an extremely high concentration of the purified total IgG from VITT patients was injected into the murine model in vivo (250 µg/g of the mouse). In addition, several ex vivo and in vitro experiments use a very high concentration of purified total IgG from 3mg/mL up to 5mg/mL, and inhibitions with IV.3 work at much lower concentrations. It is difficult to conclude the biological relevance of using very high concentrations of total IgG from VITT patients in both ex vivo and in vivo experiments described in the manuscript. Critically, no attempts have been made to quantify specific anti-PF4 IgG concentrations or their typing in the patient sera, or the protein G purified total IgG. Intriguingly in the same experiments, inhibition of FcγRIIa with deglycosylated IV.3 in vivo is performed with 1 µg/g.

We thank the reviewer for the helpful comment. We have now quantified specific anti-PF4 IgG concentrations in our total VITT IgG samples. Briefly, PF4 purified from human platelets was biotinylated and coupled to streptavidin magnetic beads according to manufacturer's instructions (New England Biolabs, Massachusetts). Purified total VITT IgG was incubated with the streptavidin conjugated PF4 mixture for 90 min at 37°C under gentle rotation. Using a magnetic separation rack, the undesired sample fraction was discarded. The retained PF4-specific VITT IgG was then eluted from the PF4-magnetic beads using acidic elution buffer (0.1M glycine) followed by immediate neutralisation using 1M Tris.

PF4-specific IgG was detected by SDS gel electrophoresis (Suppl. Fig. 2b) and quantitated by densitometry. Similar to HIT where PF4-heparin specific antibodies which constitute a small percentage of total IgG (0.8-2.2%)³¹, only a small percentage of PF4-specific antibody is present in total VITT IgG. We found that PF4-specific IgG makes up approximately 0.1-1% of total IgG. Our experience with patients with VITT, HIT and related disorders, the proportion of specific antibodies varies from patient to patient and also with timing of blood sample collection. However, please note that some patient samples were collected after treatment with immunoglobulins (IVIg) which could in part also be responsible for the variation in PF4-specific IgG present in our samples (0.1-0.3% compared to untreated samples 1%).

The activity of anti-PF4 specific IgG from VITT patients was confirmed by ELISA, SRA and flow cytometry (Suppl. Fig. 2c-e). Similar results were obtained when neutrophils were treated with 5mg/mL of total IgG (Fig. 2b) or 0.25mg/mL of PF4-specific IgG (20x less, Suppl. Fig. 2e). Based on these very small amounts of specific IgG and large amounts of non-specific IgG, the authors consider the concentration of total IgG 3-5mg/mL for in vitro experiments and 250µg/g for in vivo experiments to be appropriate and physiologically relevant.

The IV.3 monoclonal antibody by its biological nature contains only specific antibody whereas IgG from patient or control plasma are polyclonal and contain very small amount of specific antibody and large amounts of antibodies of non-related specificities. In this regard, it is not appropriate to compare monoclonal antibody (IV.3) with polyclonal antibodies from patient/control plasma. Concentrations of 1 µg/g of IV.3 have been used effectively by us¹¹ and others³².

3. Microfluidic whole blood experiments and in vivo murine model: Did the total IgG isolated from VITT patient sera produce NETosis under shear in the absence of added PF4? What is the endogenous concentration of PF4 FcγRIIa+/hPF4+ murine model under healthy conditions (i.e., non-inflammation)? Is it comparable to a healthy individual since the total IgG purified from VITT patients appears to induce platelet aggregation in PRP and NETosis in the absence of added PF4 throughout the experiments presented in the manuscript?

NETosis in microfluidics assays (under shear) was observed without the addition of extra PF4. PF4 present in plasma of donor blood was found to be sufficient for VITT IgG to form an immune complex, bind to and activate platelets and neutrophils. Similarly, VITT IgG induced platelet aggregation in PRP (which contains endogenous PF4) without the addition of exogenous PF4. As shown in Fig. 1, VITT serum can induce platelet activation (using endogenous PF4 present in serum) as determined by SRA, but activation is enhanced with the addition of exogenous PF4.

Endogenous concentration of human PF4 in the FcγRIIa/hPF4 murine model under healthy conditions was determined by ELISA. It was found that PF4 concentrations in FcγRIIa/hPF4 mice were comparable to healthy individual/human plasma levels. Wild type C57/Bl6 mice were also included as a control. As expected, there was no human PF4 expression in the C57/Bl6 mice. These data are shown in Supplementary Figure 5a (FcγRIIa/hPF4 mice: average 1113 ng/mL, range 81-3034 ng/mL, n=15; human: average 1574 ng/mL, range 544-3038 ng/mL, n=10. $P=0.07$).

4. General comment: It is kindly advised to report the detailed protocol of each experimental method as per journal policy and for replication studies.

E.g., GSK484 inhibition: Please specify the concentration used across different experiments, PF4 and PF4/H EIA studies, imaging studies.

We thank the reviewer for pointing this out. We have now added the specific concentrations of GSK484 (2μg/g) in vivo, PF4 (15 μg/ml PF4) and PF4/heparin (15 μg/mL PF4, 0.1 U/mL heparin) in the methods section. Additional concentrations were added for timelapse experiments (Sytox green (0.3μM) and Hoechst 33342 (5 ug/mL)) and mouse experiments (anti-CD42c (1ug/g), fibrinogen Alexa 594 (4ug/g), anti-citH3 Alexa 594 (0.3ug/g, abcam), anti-Ly6G Alexa 488 (0.5ug/g).

5. Discussion: anti-PF4 antibodies are frequent and it is a decade old concept in HIT that these antibodies which usually do not activate platelets are clinically irrelevant. For the same reason the presence of anti-PF4 antibodies in individuals without thrombosis had not created doubts about the concept of VITT, at least to the knowledge of this reviewer. The authors may consider to revise this part of the discussion.

Thank you for these suggestions. The authors accept and respect the reviewer's points of view. VITT is a public health issue as well as a clinical medicine issue that has attracted attention of a wider audience than HIT. The non-HIT enthusiasts may not be aware of non-pathogenic HIT antibodies. We believe it is relevant to discuss similarities and differences between HIT and VITT and these would be of interest to HIT enthusiasts and non-enthusiasts. In HIT non-pathogenic anti-PF4 antibodies are commonly found,³³ but non-activating anti-PF4 antibodies are more frequently found following COVID-19 vaccinations (both mRNA- and adenovirus-based vaccines)³⁴. Conditions like VITT and HIT are rare and

only develop in a very small subset of patients receiving heparin or COVID-19 vaccines. This has now been added to the discussion section.

References

- 1 Masuda, S. *et al.* NETosis markers: Quest for specific, objective, and quantitative markers. *Clin Chim Acta* **459**, 89-93 (2016).
- 2 Martinod, K. SSC Subcommittee Collaborative Project: Towards standardization of Neutrophil Extracellular Trap (NET) measurements in patient samples. *International Society of Thrombosis and Haemostasis*, (2019).
- 3 Koupenova, M. *et al.* The role of platelets in mediating a response to human influenza infection. *Nat Commun* **10**, 1780 (2019).
- 4 Greinacher, A. *et al.* Insights in ChAdOx1 nCoV-19 vaccine-induced immune thrombotic thrombocytopenia. *Blood* **138**, 2256-2268 (2021).
- 5 Thålin, C. *et al.* Quantification of citrullinated histones: Development of an improved assay to reliably quantify nucleosomal H3Cit in human plasma. *J Thromb Haemost* **18**, 2732-2743 (2020).
- 6 Nakayama, F. *et al.* CD15 expression in mature granulocytes is determined by alpha 1,3-fucosyltransferase IX, but in promyelocytes and monocytes by alpha 1,3-fucosyltransferase IV. *J Biol Chem* **276**, 16100-16106 (2001).
- 7 Brandau, S., Moses, K. & Lang, S. The kinship of neutrophils and granulocytic myeloid-derived suppressor cells in cancer: cousins, siblings or twins? *Semin Cancer Biol* **23**, 171-182 (2013).
- 8 Wagner, C. *et al.* Expression patterns of the lipopolysaccharide receptor CD14, and the FCgamma receptors CD16 and CD64 on polymorphonuclear neutrophils: data from patients with severe bacterial infections and lipopolysaccharide-exposed cells. *Shock* **19**, 5-12 (2003).
- 9 Gustafson, M. *et al.* A Method for Identification and Analysis of Non-Overlapping Myeloid Immunophenotypes in Humans. *PLoS ONE* **10**, e0121546 (2015).
- 10 Lee, K. *et al.* Quantification of NETs-associated markers by flow cytometry and serum assays in patients with thrombosis and sepsis. *Int J Lab Hematol* **40**, 392-399 (2018).
- 11 Perdomo, J. *et al.* Neutrophil activation and NETosis are the major drivers of thrombosis in heparin-induced thrombocytopenia. *Nat Commun* **10**, 1322 (2019).
- 12 Gavillet, M. *et al.* Flow cytometric assay for direct quantification of Neutrophil Extracellular Traps in blood samples. *Am J Hematol* **90**, 1155-1158 (2015).
- 13 Morel-Kopp, M.-C. *et al.* Heparin-induced multi-electrode aggregometry method for heparin-induced thrombocytopenia testing: communication from the SSC of the ISTH. *J Thromb Haemost* **14**, 2548-2552 (2016).
- 14 Williams, V. *et al.* Microfluidic enabling platform for cell-based assays. *J Assoc Lab Automation* **7**, 135-141 (2002).
- 15 Dupuy, A. *et al.* Thromboinflammation Model-on-A-Chip by Whole Blood Microfluidics on Fixed Human Endothelium. *Diagnostics* **11**, 203 (2021).
- 16 Fuchs, T., Brill, A., Duerschmied, D. & Wagner, C. Extracellular DNA traps promote thrombosis. *Proc Natl Acad Sci U S A* **107**, 15880-15885 (2010).
- 17 Martinod, K. & Wagner, C. Thrombosis: tangled up in NETs. *Blood* **123**, 2768-2776 (2014).
- 18 Middleton, E. *et al.* Neutrophil extracellular traps contribute to immunothrombosis in COVID-19 acute respiratory distress syndrome. *Blood* **136**, 1169-1179 (2020).
- 19 Duler, L., Nguyen, N., Ontiveros, E. & Li, R. Identification of Neutrophil Extracellular Traps in Paraffin-Embedded Feline Arterial Thrombi using Immunofluorescence Microscopy. *J Vis Exp* **157** (2020).
- 20 Arumugam, S., Subbiah, K., Kemparaju, K. & Thirunavukkarasu, C. Neutrophil extracellular traps in acrolein promoted hepatic ischemia reperfusion injury:

- Therapeutic potential of NOX2 and p38MAPK inhibitors. *J Cell Physiol* **233**, 3244-3261 (2018).
- 21 Aldabbous, L. *et al.* Neutrophil Extracellular Traps Promote Angiogenesis: Evidence From Vascular Pathology in Pulmonary Hypertension. *Arterioscler Thromb Vasc Biol* **36**, 2078-2087 (2016).
- 22 Savchenko, A. *et al.* Neutrophil extracellular traps form predominantly during the organizing stage of human venous thromboembolism development. *J Thromb Haemost* **12**, 860-870 (2014).
- 23 Shi, Y. *et al.* Neutrophils can promote clotting via FXI and impact clot structure via neutrophil extracellular traps in a distinctive manner in vitro. *Sci Rep* **11**, 1718 (2021).
- 24 Jing, C. *et al.* Antibodies Against *Pseudomonas aeruginosa* Alkaline Protease Directly Enhance Disruption of Neutrophil Extracellular Traps Mediated by This Enzyme. *Front Immunol* **12**, 654649 (2021).
- 25 Oishi, S. *et al.* Heme activates platelets and exacerbates rhabdomyolysis-induced acute kidney injury via CLEC-2 and GPVI/FcR γ . *Blood Adv* **5**, 2017-2026 (2021).
- 26 Munir, H. *et al.* Stromal-driven and Amyloid β -dependent induction of neutrophil extracellular traps modulates tumor growth. *Nat Commun* **12**, 683 (2021).
- 27 Holm, S. *et al.* Immune complexes, innate immunity, and NETosis in ChAdOx1 vaccine-induced thrombocytopenia. *European Heart Journal*, doi:10.1093/eurheartj/ehab506 (2021).
- 28 Chilingaryan, Z. *et al.* Erythrocyte interaction with neutrophil extracellular traps in coronary artery thrombosis following myocardial infarction. *Pathology* **43**, 87-94 (2022).
- 29 Leung, H. H. L. *et al.* Inhibition of NADPH oxidase blocks NETosis and reduces thrombosis in heparin-induced thrombocytopenia. *Blood Adv* **5**, 5439-5441 (2021).
- 30 Ledford, H. COVID vaccines and blood clots: what researchers know so far. *Nature* **596**, 479-481 (2021).
- 31 Suh, J., Aster, R. & Visentin, G. Antibodies from patients with heparin-induced thrombocytopenia/thrombosis recognize different epitopes on heparin: platelet factor 4. *Blood* **91**, 916-922 (1998).
- 32 Beutier, H. *et al.* Platelets expressing IgG receptor Fc γ RIIA/CD32A determine the severity of experimental anaphylaxis. *Sci Immunol* **3**, ean5997 (2018).
- 33 Warkentin, T. Challenges in Detecting Clinically Relevant Heparin-Induced Thrombocytopenia Antibodies. *Hamostaseologie* **40**, 472-484 (2020).
- 34 Thiele, T. *et al.* Frequency of positive anti-PF4/polyanion antibody tests after COVID-19 vaccination with ChAdOx1 nCoV-19 and BNT162b2. *Blood* **138**, 299-303 (2021).

Reviewers' Comments:

Reviewer #1:

Remarks to the Author:

It is commendable that the authors have considered the comments carefully, and addressed most of them with additional experiments and further verifications. The following minor corrections are required to improve the quality of the manuscript:

1- Supplemental figure 1d: Indicate the NET% in the upper right quadrant, similar to 1e. Provide the details as to whether these experiments were done with fresh blood from healthy, vaccinated or patient samples in the fig legends and in the results.

2- Page 5; lines 1-5 indicates that monocytes also induce extracellular trap formation. It is important to show the n-values and the statistical significance level to make such conclusions. Furthermore, if monocytes are also releasing ETs during VITT, contributions of METosis need to be highlighted with additional data for monocyte/macrophage staining in the tissues. If the clarification experiments are not feasible, the data need to be interpreted and discussed cautiously, considering that the title and abstract talk about NETosis, not METosis in VITT.

3- Supplemental figure 2 b: Indicate the MW of the standard bands, and the band corresponding to the arrow. Since only one band is visible for the antibodies, clarify whether these are non-reducing gels, in the fig legend.

4- Either remove "data not shown" or show the data in the supplemental section.

5- Specific type of NETosis induced by ligand-platelet-neutrophil interactions has been shown in 2007 for LPS/bacteria as a unique form of NETosis (e.g., Nat Med. 2007 Apr;13(4):463-9. doi: 10.1038/nm1565). Pathway engaged by immune complexes engagement with Fc receptors has also been elaborated (e.g., J Clin Invest. 2017 Oct 2;127(10):3810-3826. doi: 10.1172/JCI94039). Furthermore, PAD-4-mediated CitH3 formation is a calcium-dependent process; and the activation of ROS by NADPH is not necessary e.g., (Proc Natl Acad Sci U S A. 2015 Mar 3;112(9):2817-22. doi: 10.1073/pnas.1414055112). These points are worth discussing, since the authors are unable to conduct the pathway-specific mechanistic experiments for this paper.

Reviewer #2:

Remarks to the Author:

The authors have responded to my concerns satisfactorily. The revised manuscript is improved. I have no further issues.

Reviewer #3:

Remarks to the Author:

The authors addressed all comments of the reviewers and performed relevant additional experiments. Overall the manuscript is well written and contributes substantial and relevant new information.

Minor comments

Page 3, line 27: the statement that there is a lack of scientific evidence that VITT antibodies induce clot formation in vitro is incorrect. Flow chamber experiments showing clot formation and involvement of platelets and granulocytes have also been published by others.

<https://doi.org/10.3324/haematol.2021.280251>

page 3, line 45: "we confirm and extend" would be more appropriate than "we show" for the first part, while the in vivo experiments are new.

Page 7, line 13: "despite no direct evidence"; please see comment above

REVIEWERS' COMMENTS

Reviewer #1 (Remarks to the Author):

It is commendable that the authors have considered the comments carefully, and addressed most of them with additional experiments and further verifications. The following minor corrections are required to improve the quality of the manuscript:

1- Supplemental figure 1d: Indicate the NET% in the upper right quadrant, similar to 1e. Provide the details as to whether these experiments were done with fresh blood from healthy, vaccinated or patient samples in the fig legends and in the results.

The authors thank Reviewer #1 for the reviewer's effort. These are our replies to the comments.

Supplemental Figure 1d exemplifies the gating strategy used to identify PNA and NETs. The authors included the gating strategy (Suppl. Fig. 1a-d) in the Supplemental Information to adhere to the Nature portfolio reporting checklist requirements. We have now included the NET% in the upper right quadrant (as requested by the reviewer). The figure legend has been updated to specify the blood used in this example gating strategy.

2- Page 5; lines 1-5 indicates that monocytes also induce extracellular trap formation. It is important to show the n-values and the statistical significance level to make such conclusions. Furthermore, if monocytes are also releasing ETs during VITT, contributions of METosis need to be highlighted with additional data for monocyte/macrophage staining in the tissues. If the clarification experiments are not feasible, the data need to be interpreted and discussed cautiously, considering that the title and abstract talk about NETosis, not METosis in VITT.

Statistical analysis of data acquired from Suppl. Fig. 1e are now shown in Suppl Fig. 1f. The n-values and statistical significance have been added to the figure legend. Data on monocyte ETs presented were among results collected from previous experiments; no new experiments were performed. The authors have also clarified in the manuscript that although a low percentage of monocytes can be activated by VITT IgG, monocytes are likely to contribute minimally to overall ETs due to their very small numbers in circulation and much lower reactivity to VITT IgG compared to neutrophils.

Clarification experiments are not feasible as monocytes are so few in number and based on our results (Suppl Fig. 1e, f) only a very small percentage is likely to be activated. Finding CitH3 in or near monocytes will be unlikely and would require additional experiments beyond the scope of this paper. Additionally, positive staining of CitH3 near a monocyte does not confirm that CitH3 was derived from monocytes. Hence the finding, if successfully obtained, is inconclusive and would not add scientifically to the data already presented.

Our data show that ETs are predominantly derived from neutrophils rather than monocytes. Hence in VITT, NETosis is the main contributor to thrombosis as highlighted in our title and paper.

3- Supplemental figure 2 b: Indicate the MW of the standard bands, and the band corresponding to the arrow. Since only one band is visible for the antibodies, clarify whether these are non-reducing gels, in the fig legend.

The authors thank the reviewer for pointing this out. We have now updated Suppl. Fig. 2b with the MW of the standard bands and the band corresponding to the arrow. The figure legend has been

revised to specify that the SDS PAGE was under non-reducing conditions.

4- Either remove “data not shown” or show the data in the supplemental section.

The authors have now removed “data not shown”.

5- Specific type of NETosis induced by ligand-platelet-neutrophil interactions has been shown in 2007 for LPS/bacteria as a unique form of NETosis (e.g., Nat Med. 2007 Apr;13(4):463-9. doi: 10.1038/nm1565). Pathway engaged by immune complexes engagement with Fc receptors has also been elaborated (e.g., J Clin Invest. 2017 Oct 2;127(10):3810-3826. doi: 10.1172/JCI94039). Furthermore, PAD-4-mediated CitH3 formation is a calcium-dependent process; and the activation of ROS by NADPH is not necessary e.g., (Proc Natl Acad Sci U S A. 2015 Mar 3;112(9):2817-22. doi: 10.1073/pnas.1414055112). These points are worth discussing, since the authors are unable to conduct the pathway-specific mechanistic experiments for this paper.

The authors agree that there are various pathway-specific mechanisms involved in different types of NETosis and these NET requirements are agonist-specific (Kenny et al., eLife 2017; Yipp et al., Blood 2013). For example, nicotinamide adenine dinucleotide phosphate (NADPH) oxidase (NOX) activity is required for phorbol 12-myristate 13-acetate (PMA) and some bacteria-induced NETs but is dispensable for NET formation in ionophore-stimulated neutrophils (Kenny et al., eLife 2017). We have also shown that ROS and NOX2 play a crucial role in NETosis and thrombosis in HIT (Leung et al., Blood Adv 2021). Due to the analogous nature of HIT and VIT, we expect that NET formation is likely dependent on both PAD4 and NADPH oxidase, as previously shown in HIT. This is now discussed in the manuscript.

NETosis mechanisms differ based on their inciting stimuli and timing of NET release. Vital NETosis, activated by microbial pathogens and lipopolysaccharide (LPS), induces rapid NET release via nuclear budding and vesicular release, sparing the outer membrane. Conversely, suicidal NETosis involves the decondensation of chromatin, membrane rupture and extrusion of NETs from the neutrophil. Since infection-induced NETs differs from immune complex-induced NETs, vital NETosis is not considered in the manuscript.

Reviewer #2 (Remarks to the Author):

The authors have responded to my concerns satisfactorily. The revised manuscript is improved. I have no further issues.

The authors thank Reviewer #2 for the reviewer’s time and comments.

Reviewer #3 (Remarks to the Author):

The authors addressed all comments of the reviewers and performed relevant additional experiments. Overall the manuscript is well written and contributes substantial and relevant new information.

Minor comments

Page 3, line 27: the statement that there is a lack of scientific evidence that VITT antibodies induce clot formation in vitro is incorrect. Flow chamber experiments showing clot formation and involvement of platelets and granulocytes have also been published by others.

<https://doi.org/10.3324/haematol.2021.280251>

The authors thank Reviewer #3 for their time and helpful comments. When we first submitted this paper (7 September 2021), there were no published findings that VITT antibodies induce clot formation in vitro.

The preprint of this *Nature Communications* manuscript was published in September 2021, prior to the submission of the reviewer's referenced article to *Haematologica* in November 2021. Therefore, the referenced paper confirms our findings, which have precedence over the other paper.

As outlined in the Nature Communications policy (*Nat Comm* **11**, 4466 (2020)), competing work published while a manuscript is under review at *Nature Communications* will not compromise the novelty of the study.

Hence, our preferred option is not to change our wording if the editor agrees.

If the editor does not accept our precedence of these findings, the authors would agree to reword the statement to: subsequent to our submission to the journal, the induction of thrombosis by VITT plasma in vitro was reported in another journal, with a reference to the paper.

page 3, line 45: "we confirm and extend" would be more appropriate than "we show" for the first part, while the in vivo experiments are new.

In this sentence, 'show' is used to describe and summarise the findings in this manuscript. The authors believe that the statement "we show" does not infer precedence per se, and does not need to be changed.

Page 7, line 13: "despite no direct evidence"; please see comment above
Please see response above regarding precedence of our submission and preprint.